# Safe Linear-Quadratic Dual Control with Almost Sure Performance Guarantee

## ABSTRACT

This paper considers the linear-quadratic dual control problem where the system parameters need to be identified and the control objective needs to be optimized in the meantime. Contrary to existing works on data-driven linear-quadratic regulation, which typically provide error or regret bounds within a certain probability, we propose an online algorithm that guarantees the asymptotic optimality of the controller in the almost sure sense. Our dual control strategy consists of two parts: a switched controller with time-decaying exploration noise and Markov parameter inference based on the cross-correlation between the exploration noise and system output. Central to the almost sure performance guarantee is a safe switched control strategy that falls back to a known conservative but stable controller when the actual state deviates significantly from the target state. We prove that this switching strategy rules out any potential destabilizing controllers from being applied, while the performance gap between our switching strategy and the optimal linear state feedback is exponentially small. Under our dual control scheme, the parameter inference error scales as $\mathcal{O}(T^{-1/4+\epsilon})$, while the suboptimality gap of control performance scales as $\mathcal{O}(T^{-1/2+\epsilon})$, where $T$ is the number of time steps, and $\epsilon$ is an arbitrarily small positive number. Simulation results on an industrial process example are provided to illustrate the effectiveness of our proposed strategy.

## 1 INTRODUCTION

One of the most fundamental and well-studied problems in optimal control, Linear-Quadratic Regulation (LQR) has recently aroused renewed interest in the context of data-driven control and reinforcement learning. Considering it is usually challenging to obtain an exact system model from first principles, and that the system may slowly change over time due to various reasons, e.g., component wear-out, data-driven regulation of unknown linear systems has become an active research problem in the intersection of machine learning and control, with recent works including e.g., Dean et al. (2019); Mania et al. (2019); Cohen et al. (2019); Wagenmaker & Jamieson (2020). In particular, from the perspective of reinforcement learning theory, the LQR problem has become a standard benchmark for continuous control.

In this paper, we focus on the dual control (Feldbaum, 1960) setting, also known as online adaptive control in the literature, where the same policy must be adopted to identify the system parameters and optimize the control objective, leading to the well-known exploration-exploitation dilemma. Recently, it was shown by Simchowitz & Foster (2020) that the optimal regret for this problem setting scales as $\tilde{\Theta}(\sqrt{T})$, which can be achieved with probability $1 - \delta$ using a certainty equivalent control strategy, where the learner selects control inputs according to the optimal controller for the current estimate of the system while injecting time-decaying exploration noise. However, the strategy proposed in this work, like those in its predecessors (Abbasi-Yadkori & Szepesvári, 2011; Dean et al., 2018; Mania et al., 2019), may have a nonzero probability $\delta$ of failing. Furthermore, it shall be noticed that $\delta$ has been chosen as a fixed design parameter in the aforementioned works, which implies the probability of failing does not converge to zero even if the policy is run indefinitely. The above observation gives rise to the question that we address in this paper:

> Can we design a learning scheme for LQR dual control, such that the policy
> adopted in *almost every* trajectory converges to the optimal policy?

We identify that the above goal can hardly be achieved by a naive certainty equivalent learning scheme. Qualitatively, the system parameters learned from data are always corrupted by random noise, and as a result, the controller proposed in previous works may destabilize the system, albeit with a small probability, causing catastrophic system failure. Based on the above reasoning, we propose a notion of bounded-cost safety for the LQR dual control problem: we recognize a learning scheme to be safe if no destabilizing control policy is applied during the entire learning process.

In this paper, we propose a learning scheme that satisfies the above definition of bounded-cost safety, and guarantees both the parameter inference error and the suboptimality gap of controller performance converge to zero almost surely. Our strategy consists of two parts: a safe switched controller and a parameter inference algorithm. The switched controller can be viewed as a safety-augmented version of the certainty equivalent controller: it normally selects control inputs according to the optimal linear feedback for the currently estimated system parameters, but falls back to a conservative controller for several steps when the actual state deviates significantly from the target state. We prove that this switching strategy ensures the bounded-cost safety of the learning process, while only inducing a suboptimality gap that decays exponentially as the switching threshold increases. For the parameter inference part, in contrast to the direct least-squares approach for estimating the matrices $A, B$ widely adopted in the literature, we estimate the Markov parameters, also known as the impulse response of the system, based on the cross-correlation between the exploration noise and the system output, and establish the almost-sure convergence using a law of large numbers for martingales. We prefer this approach for its clarity in physical meaning and simplicity of convergence analysis, but we do not foresee substantial difficulty in replacing our parameter inference module with standard least-squares. We prove that under the above described learning scheme, the parameter inference error scales as $\mathcal{O}(T^{-1/4+\epsilon})$, while the suboptimality gap of control performance scales as $\mathcal{O}(T^{-1/2+\epsilon})$, where $T$ is the number of time steps, and $\epsilon$ is an arbitrarily small positive number. Both the above results match the corresponding asymptotic rates in Simchowitz & Foster (2020), which provides an rate-optimal algorithm for online LQR in the high-probability regime.

The main contributions of this paper are as follows:

1. We propose a practical notion of safety for the LQR dual control problem that has not been considered in the literature, and provide an instance of safe learning scheme based on a switching strategy.

2. We prove almost sure convergence rates of parameter inference error and suboptimality gap of control performance for our scheme, which match the corresponding optimal rates in the high-probability regime. To the best of our knowledge, this is the first analysis of the almost sure convergence rate for online LQR.

The rest of this paper is organized as follows: Section 2 gives a brief introduction of LQR, formulates the LQR dual control problem, and defines the performance metrics as well as the notion of bounded-cost safe learning scheme. Section 3 presents and interprets our algorithm. Section 4 states the main theoretical results and characterizes the convergence rates. Section 5 provides simulation results on an industrial process example to illustrate the effectiveness of our proposed strategy. Section 6 summarizes the related literature. Finally, Section 7 gives concluding remarks and discusses future directions.

## 2 PROBLEM FORMULATION

We consider the control of the following discrete-time linear system:

$$x_{k+1} = Ax_k + Bu_k + w_k, \tag{1}$$

where $x_k \in \mathbb{R}^n$ is the state vector, $u_k \in \mathbb{R}^p$ is the input vector, and $w_k \in \mathbb{R}^n$ is the process noise. We assume $x_0 \sim \mathcal{N}(0, X_0)$, $w_k \sim \mathcal{N}(0, W)$, and that $x_0, w_0, w_1, \ldots$ are pairwise independent. We also assume w.l.o.g. that $(A, B)$ is controllable.

We consider control policies of the form

$$\pi : \mathbb{R}^n \times \mathbb{R}^q \to \mathbb{R}^p \times \mathbb{R}^q, (u_k, \xi_{k+1}) = \pi(x_k, \xi_k), \tag{2}$$

which can be either deterministic or stochastic, where $\xi_k \in \mathbb{R}^q$ is the internal state of the policy. Notice that we allow the flexibility of the policy being non-Markovian with the introduction of $\xi_k$,

and we also use the simplified notation $u_k = \pi(x_k)$ if $\pi$ is Markovian. The performance of a policy $\pi$ can be characterized by the infinite-horizon quadratic cost

$$J^\pi = \limsup_{T \to \infty} \frac{1}{T} \mathbb{E} \left[ \sum_{k=0}^{T-1} x_k^\top Q x_k + u_k^\top R u_k \right], \qquad (3)$$

where $Q \succ 0, R \succ 0$ are known weight matrices specified by the system operator.

We denote the optimal cost and the optimal control law by

$$J^* = \inf_\pi J^\pi, \pi^* \in \arg\min_\pi J^\pi. \qquad (4)$$

It is well-known that the optimal policy is a linear function of the state $\pi^*(x) = K^* x$, with associated cost $J^* = \operatorname{tr}(W P^*)$, where $P^*$ is the solution to the discrete-time algebraic Riccati equation

$$P^* = Q + A^\top P^* A - A^\top P^* B \left( R + B^\top P^* B \right)^{-1} B^\top P^* A, \qquad (5)$$

and the linear feedback control gain $K^*$ can be determined by

$$K^* = - \left( R + B^\top P B \right)^{-1} B^\top P^* A. \qquad (6)$$

Based on the definitions above, we can use $J^\pi - J^*$ to measure of the suboptimality gap of a specific policy $\pi$.

In the online LQR setting, the system and input matrices $A, B$ are assumed unknown, and the learning process can be viewed as deploying a sequence of time-varying policies $\{\pi_k\}$ with the dual objectives of exploring the system parameters and stabilizing the system. To characterize the safety of a learning process, we make the definitions below:

**Definition 1.** *A policy $\pi$ is destabilizing if $J^\pi = +\infty$.*

**Definition 2.** *A learning process applying policies $\{\pi_k\}_{k=0}^\infty$ is bounded-cost safe, if $\pi_k$ is not destabilizing for any time $k$ and for any realization of the noise process.*

Notice that Definition 1 is a generalization of the common notion of destabilizing linear feedback gain, i.e., $\pi(x) = Kx$ is destabilizing when $\rho(A + BK) \geq 1$, which is equivalent to $J^\pi = +\infty$. Based on the notion of destabilizing policies, we propose the concept of bounded-cost safety in Definition 2, which requires that destabilizing policies, or policies with unbounded cost, are never applied. It should be pointed out that bounded-cost safety does not guarantee the stability of trajectories, but is an indicator of the reliability of a learning scheme.

We assume the system is open-loop strictly stable, i.e., $\rho(A) < 1$. Indeed, if there is a known stabilizing linear feedback gain $K_0$, then we can apply the dual control scheme on the pre-stabilized system $(A + BK_0, B)$ instead of $(A, B)$. Existence of such a known stabilizing linear feedback gain a standard assumption in previous works on online LQR (Mania et al., 2019; Simchowitz & Foster, 2020), and relatively easy to establish through coarse system identification (Dean et al., 2019; Faradonbeh et al., 2018b) or adaptive stabilization methods (Faradonbeh et al., 2018a; 2019).

## 3 ALGORITHM

The complete algorithm we propose for LQR dual control is presented in Algorithm 1. The modules in the algorithm will be described later in this section.

### 3.1 SAFE CONTROL POLICY

This subsection describes the policy for determining the control input $u_k$. The first $n + p$ steps are a warm-up period where purely random inputs are injected. Afterwards, in each step, we inject an exploitation term $\tilde{u}$ plus a polynomial-decaying exploratory noise $(k+1)^{-\beta}\zeta$, where the decay rate $\beta \in (0, 1/2)$ is a constant. The exploitation term $\tilde{u}$ is a modified version of the certainty equivalent control input $\hat{K}_k x_k$, and this modification, described in Algorithm 2, is crucial to the safety of the learning process, which we will detail below.

---

**Algorithm 1** Safe LQR dual control

---

**Input:** State dimension $n$, input dimension $p$, exploratory noise decay rate $\beta$

1: **for** $k = 0, 1, \ldots, n + p - 1$ **do**
2: $\quad \xi_{k+1} \leftarrow 0$
3: $\quad$ Apply control input $u_k \leftarrow (k+1)^{-\beta} \zeta_k$, where $\zeta_k \sim \mathcal{N}(0, I_p)$
4: **for** $k = n + p, n + p + 1, \ldots$ **do**
5: $\quad$ Observe the current state $x_k$
6: $\quad$ **for** $\tau = 0, 1, \ldots, n + p - 1$ **do**
7:
$$\hat{H}_{k,\tau} \leftarrow \frac{1}{k - \tau} \sum_{i=\tau+1}^{k} (i - \tau)^{\beta} \left[ x_i - \sum_{t=0}^{\tau-1} \hat{H}_{k,t} \tilde{u}_{i-t-1} \right] \zeta_{i-\tau-1}^{\top}$$
8: $\quad$ Reconstruct $\hat{A}_k, \hat{B}_k$ from $\hat{H}_{k,0}, \ldots, \hat{H}_{k,n+p-1}$ using Algorithm 3
9: $\quad$ Compute certainty equivalent feedback gain $\hat{K}_k$ by replacing $A, B$ with $\hat{A}_k, \hat{B}_k$ in (5),(6)
10: $\quad$ Determine policy $\pi_k(\cdot, \cdot) \leftarrow \pi(\cdot, \cdot; k, \hat{K}_k, \beta)$, where $\pi$ is described by Algorithm 2
11: $\quad (u_k, \xi_{k+1}) \leftarrow \pi_k(x_k, \xi_k)$; record $\tilde{u}_k \leftarrow \tilde{u}, \zeta_k \leftarrow \zeta$, where $\tilde{u}, \zeta$ are the corresponding variables generated when executing the policy
12: $\quad$ Apply control input $u_k$

---

**Algorithm 2** Safe policy $\pi(x, \xi; k, K, \beta)$

---

**Input:** Arguments: system state $x$, policy internal state $\xi$; Parameters: step $k$, linear feedback gain $K$, exploratory noise decay rate $\beta$
**Output:** Control input $u$ and next policy internal state $\xi'$, i.e., $(u, \xi') = \pi(x, \xi; k, K, \beta)$

1: **if** $\xi > 0$ **then**
2: $\quad \tilde{u} \leftarrow 0, \xi' \leftarrow \xi - 1$
3: **else**
4: $\quad$ **if** $\max\{\|K\|, \|x\|\} \geq \log k$ **then**
5: $\quad\quad \tilde{u} \leftarrow 0, \xi' \leftarrow \lfloor \log k \rfloor$
6: $\quad$ **else**
7: $\quad\quad \tilde{u} \leftarrow Kx, \xi' \leftarrow 0$
8: $u \leftarrow \tilde{u} + (k+1)^{-\beta} \zeta$, where $\zeta \sim \mathcal{N}(0, I)$

---

In short, we stop injecting the exploitation input for $\lfloor \log k \rfloor + 1$ consecutive steps, if either the state norm $\|x_k\|$ or the norm of the feedback gain $\|\hat{K}_k\|$ exceeds the threshold $\log k$. Recall from (2) that we use $\xi_k$ to denote the internal state of the policy, and in Algorithm 1, $\xi_k$ is a counter that records how many steps are left in the "non-action" period. Essentially, we utilize the innate stability of the system to prevent the state from exploding catastrophically. This "non-action" mechanism is a critical feature of our control design, without which the controller learned from data may destabilize the system, albeit with a small probability, causing system failure in practice and forbidding the establishment of almost sure performance guarantees in theory. We provide an ablation study of this "non-action" mechanism in Section 5.

We choose both the switching threshold and the length of the "non-action" period to be time-growing. The enlarging threshold corresponds to diminishing degree of conservativeness, which is essential for the policy performance to converge to the optimal performance. Meanwhile, the prolonging "non-action" period rules out the potential oscillation of state caused by the frequent switching of the controller (see Appendix B for an illustrative example of this oscillation phenomenon). In particular, it can be shown that the suboptimality gap incurred by the switching strategy scales as $\mathcal{O}(tM \exp(-cM^2))$, where $M$ is the switching threshold, $t$ is the length of the "non-action" period, and $c$ is a system-dependent constant (see Lemma 10 in Appendix A.1.2). With both $M$ and $t$ growing as $\mathcal{O}(\log k)$, the contribution of our switching strategy to the overall suboptimality gap is merely $\tilde{\mathcal{O}}(1)$.

## 3.2 PARAMETER INFERENCE

The Markov parameters of the system described in (1) are defined as

$$H_\tau \triangleq A^\tau B, \quad \tau = 0, 1, \dots. \tag{7}$$

The Markov parameter sequence $\{H_\tau\}_{\tau=0}^\infty$ can be interpreted as the impulse response of the system. It shall be noted that finite terms of the Markov parameter sequence would suffice to characterize the system, since higher-order Markov parameters can be represented as the linear combination of lower-order Markov parameters using the characteristic polynomial of $A$ in combination with Cayley-Hamilton theorem. In particular, our inference algorithm estimates the first $n + p$ terms of the impulse response. We denote our estimate of $H_\tau$ at step $k$ by $\hat{H}_{k,\tau}$, whose expression is specified in line 7 of Algorithm 1. Based on the cross-correlation between the current state $x_k$ and a past exploratory noise input $\zeta_{k-\tau-1}$, we can produce an unbiased estimate of $H_\tau$ at each step, and $\hat{H}_{k,\tau}$ is a cumulative average of such unbiased estimates, whose almost-sure convergence to $H_\tau$ can be established through a law of large numbers for martingales (see Appendix A.1.3).

**Remark 1.** *From a computational perspective, line 7 of Algorithm 1 can be decomposed to enable efficient recursive updates. To see this, we can rearrange this expression as*

$$\hat{H}_{k,\tau} = \frac{1}{k-\tau} \sum_{i=\tau+1}^k (i-\tau)^\beta x_i \zeta_{i-\tau-1}^\top - \sum_{t=0}^{\tau-1} \hat{H}_{k,t} \left[ \frac{1}{k-\tau} \sum_{i=\tau+1}^k (i-\tau)^\beta \tilde{u}_{i-t-1} \zeta_{i-\tau-1}^\top \right],$$

*the weighted sum of $\tau + 1$ cumulative averages, each of which can be updated with a constant amount of computation in each step. Therefore, the time complexity of each iteration and the total memory consumption are constant.*

From the inferred Markov parameters $\{\hat{H}_{k,\tau}\}_{\tau=0}^{n+p-1}$, we can reconstruct $\hat{A}_k, \hat{B}_k$ by fitting the data obtained from several rollouts on the virtual system described by $\{\hat{H}_{k,\tau}\}_{\tau=0}^{n+p-1}$. The procedure is detailed in Algorithm 3.

---

**Algorithm 3** Restoring system and input matrices from Markov parameters

---

**Input:** Markov parameter estimate sequence $\hat{H}_0, \hat{H}_1, \dots, \hat{H}_{n+p-1}$, number of simulated trajectories $N$

**Output:** System matrices estimate $\hat{A}, \hat{B}$

1: Construct block Toeplitz matrix $\mathcal{T} \leftarrow \begin{bmatrix} 0 & 0 & 0 & \cdots & 0 \\ \hat{H}_0 & 0 & 0 & \cdots & 0 \\ \hat{H}_1 & \hat{H}_0 & 0 & \cdots & 0 \\ \vdots & \vdots & \vdots & \ddots & \vdots \\ \hat{H}_{n+p-1} & \hat{H}_{n+p-2} & \hat{H}_{n+p-3} & \cdots & 0 \end{bmatrix}$

2: Choose $N$ independent random input trajectories, each $(n + p)$-long, denoted by $\mathbf{u}_i \leftarrow \begin{bmatrix} u_i^{(1)} & u_i^{(2)} & \cdots & u_i^{(N)} \end{bmatrix}$, $i = 0, 1, \dots, n + p - 1$.

3: Stack the inputs vertically and compute states: $\mathcal{U}^v \leftarrow [\mathbf{u}_0; \mathbf{u}_1; \cdots; \mathbf{u}_{n+p-1}], [\mathbf{x}_1; \mathbf{x}_2; \cdots; \mathbf{x}_{n+p}] =: \mathcal{X}_1^v \leftarrow \mathcal{T}\mathcal{U}^v$

4: Stack the inputs and states horizontally to form the following matrices: $\mathcal{U}^h \leftarrow [\mathbf{u}_0 \quad \mathbf{u}_1 \quad \cdots \quad \mathbf{u}_{n+p-1}], \mathcal{X}_1^h \leftarrow [\mathbf{x}_1 \quad \mathbf{x}_2 \quad \cdots \quad \mathbf{x}_{n+p}], \mathcal{X}_0^h \leftarrow [\mathbf{0}_{n\times N} \quad \mathbf{x}_1 \quad \cdots \quad \mathbf{x}_{n+p-1}]$

5: Compute estimate $\begin{bmatrix} \hat{B} & \hat{A} \end{bmatrix} \leftarrow \mathcal{X}_1^h \begin{bmatrix} \mathcal{U}^h; \mathcal{X}_0^h \end{bmatrix}^\dagger$.

---

It can be shown the procedure described in Algorithm 3 restores the actual system and input matrices as long as the Markov parameter estimates are accurate:

**Lemma 1.** *When $\hat{H}_\tau = H_\tau = A^\tau B$ for all $\tau = 0, 1, \dots, n + p - 1$, and the matrix $\begin{bmatrix} \mathcal{U}^h \\ \mathcal{X}_0^h \end{bmatrix}$ defined in Algorithm 3 has full row rank, the result of Algorithm 3 satisfies $\hat{A} = A, \hat{B} = B$.*

*Proof.* When $\hat{H}_\tau = A^\tau B$, we have $\mathcal{X}_1^h = A\mathcal{X}_0^h + B\mathcal{U}^h = \begin{bmatrix} B & A \end{bmatrix} \begin{bmatrix} \mathcal{U}^h \\ \mathcal{X}_0^h \end{bmatrix}$. Since $\begin{bmatrix} \mathcal{U}^h \\ \mathcal{X}_0^h \end{bmatrix}$ has full row

rank, we have $\begin{bmatrix} \mathcal{U}^h \\ \mathcal{X}_0^h \end{bmatrix} \begin{bmatrix} \mathcal{U}^h \\ \mathcal{X}_0^h \end{bmatrix}^\dagger = I$, and therefore $\begin{bmatrix} B & A \end{bmatrix} = \mathcal{X}_1^h \begin{bmatrix} \mathcal{U}^h \\ \mathcal{X}_0^h \end{bmatrix}^\dagger = \begin{bmatrix} \hat{B} & \hat{A} \end{bmatrix}$. □

**Remark 2.** *Lemma 1 guarantees the consistency of our controller, i.e., accurate Markov parameter estimates would generate an optimal exploitation input. The full-row-rank condition in Lemma 1, naturally satisfied by randomly generated inputs, guarantees that the inputs provide sufficient excitation to reconstruct system matrices from impulse responses. For the ease of analysis, it is also legitimate to use fixed, rather than random $\mathbf{u}_i$'s for each time step $k$ of the outer loop of Algorithm 1, as long as the rank condition is satisfied.*

## 4 MAIN RESULTS

The main theoretical properties of our proposed LQR dual control scheme are stated as follows:

**Theorem 1.** *Assuming $A$ is stable, the learning process described in Algorithm 1 is bounded-cost safe according to Definition 2.*

**Theorem 2.** *Assuming $A$ is stable, and $0 < \beta < 1/2$, for $\hat{H}_{k,\tau}$ computed in Algorithm 1, the following limit holds almost surely, i.e., with probability 1:*

$$\lim_{k \to \infty} \frac{\hat{H}_{k,\tau} - H_\tau}{k^{-\gamma+\epsilon}} = 0, \tag{8}$$

*where $\gamma = 1/2 - \beta > 0$, for any $\epsilon > 0$ and any $\tau = 0, 1, \ldots, n+p-1$.*

**Remark 3.** *Theorem 2 states that our estimate of the Markov parameters converge at the order $\mathcal{O}\left(k^{-\gamma+\epsilon}\right)$. As a corollary, our estimate of the system and input matrices $A, B$ also converge at $\mathcal{O}\left(k^{-\gamma+\epsilon}\right)$ (see Appendix A.4). This convergence rate coincides with the one guaranteed by the more commonly used least-squares method, for which one can establish the convergence of estimates of $A, B$ at $\mathcal{O}\left(k^{-\gamma+\epsilon}\right)$ using concentration bounds for martingale least-squares (e.g., Lemma E.1 in Simchowitz & Foster (2020)). Therefore, our parameter inference algorithm based on cross-correlation can be viewed as a competitive alternative to the least-squares estimator. One feature of our cross-correlation approach compared to the least-squares approach, however, is that the estimation of Markov parameters can be easily extended to the partially observable LQR setting like the one considered in Zheng et al. (2020). Also, the convergence of our cross-correlation approach is not based on the sub-Gaussianity of the process noise (see Appendix A.3), which allows straightforward generalization to long-tailed noise models.*

**Theorem 3.** *Assuming $A$ is stable, and $0 < \beta < 1/2$, let $\pi_k$ be the control policy used at step $k$ in Algorithm 1, then with $J^{\pi_k}$ and $J^*$ defined in (3), (4) respectively, for any $\epsilon > 0$, the following limit holds almost surely, i.e., with probability 1:*

$$\lim_{k \to \infty} \frac{J^{\pi_k} - J^*}{k^{-\min(2\beta, 2\gamma)+\epsilon}} = 0, \tag{9}$$

*where $\gamma = 1/2 - \beta > 0$.*

Due to space limits, all the proofs are deferred to the appendix.

**Remark 4.** *According to Theorem 2, the convergence rate $\gamma$ is maximized when $\beta \to 0^+$. However, the exploration term $k^{-\beta}$ does not decay in this case, and the control performance $J^{\pi_k}$ will not converge to the optimal performance $J^*$. To achieve the fastest convergence of $J^{\pi_k}$, we need to choose the decay rate of the exploration term to be $\beta = 1/4$, which maximizes $\min(2\beta, 2\gamma) = \min(2\beta, 1 - 2\beta)$. When $\beta = 1/4$, we have both parameter estimation errors and policy suboptimality gaps scaling at $\tilde{\mathcal{O}}(1/\sqrt{T})$, which matches the asymptotic rates of corresponding components in the (with-high-probability) regret-optimal algorithm proposed in Simchowitz & Foster (2020). However, due to the challenging nature of the analysis of the nonlinear closed-loop system under our safe control scheme, the exact regret of our scheme is still under investigation.*

## 5 SIMULATION

In this section, the performance of our proposed algorithm is evaluated using Tennessee Eastman Process (TEP), an industrial process example for benchmarking linear controllers (Ricker, 1993). We used a simplified version of TEP from Liu et al. (2020), where the state and input dimensions are $n = 8, p = 4$, and the spectral radius of the system matrix is $\rho(A) \approx 0.96$. We assume $Q, R$, are identity matrices, and $W, X_0$ are also identity matrices, i.e., the process noise is i.i.d. standard Gaussian. We choose $N = 50$ (in Algorithm 3) for all the experiments. In the practical implementation, the estimated system matrices $\hat{A}_k, \hat{B}_k$ and the feedback gain $\hat{K}_k$ are updated only at steps $\lfloor 10^{n/2} \rfloor (n \in \mathbb{N})$ to speed up the computation.

To illustrate the impact of the parameter $\beta$ on the convergence of the algorithm, we perform 100 independent experiments for each of $\beta \in \{0, 1/4, 1/2\}$, with $10^8$ steps in each experiment. Fig. 1 shows the error of the estimated system and input matrices, i.e., $\left\| \hat{A}_k - A \right\|$ and $\left\| \hat{B}_k - B \right\|$ against the time $k$ for different values of $\beta$.

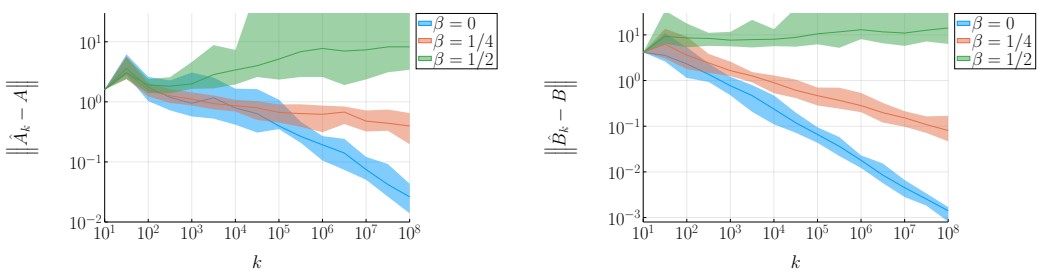

Figure 1: Error of estimated system matrices for different $\beta$ against time $k$. The solid lines are the median among the experiments, and the shades represent the range among the experiments.

From Fig. 1, one can see when $\beta$ is 0 or $1/4$, the estimation error of the system matrices converges to zero as time $k$ goes to infinity, and the convergence approximately follows a power law. Furthermore, the convergence speed of the estimation error is significantly faster when $\beta = 0$. Meanwhile, when $\beta = 1/2$, the estimate error diverges in some of the experiments. The above observations are consistent with the theoretical result in Theorem 2, where it is stated that the parameter estimates converge with rate $\mathcal{O}\left(k^{-\gamma+\epsilon}\right)$, with $\gamma = 1/2 - \beta$, i.e., convergence is only guaranteed with $\beta < 1/2$ and is faster when $\beta$ is smaller.

Now we consider the performance of policies with different values of $\beta$. To quantify the policy performance, one shall in theory compute $J^{\pi_k}$ as defined in (3). However, the policies $\pi_k$ are nonlinear, rendering it very difficult, if not impossible, to compute $J^{\pi_k}$ analytically. As a surrogate approach for evaluating a policy $\pi_k$, we define the empirical cost $\hat{J}^{\pi_k}$ as

$$\hat{J}^{\pi_k} = \frac{1}{N} \frac{1}{T} \sum_{i=1}^{N} \sum_{t=0}^{T-1} \left(x_t^{(i)}\right)^\top Q x_t^{(i)} + \left(u_t^{(i)}\right)^\top R u_t^{(i)},$$

where $x_t^{(i)}, u_t^{(i)}$ are states and inputs collected from the closed-loop system under $\pi_k$ in $N$ independent $T$-long sample paths indexed by $i$. In our simulations, we take $T = 10000, N = 10$ to evaluate each stabilizing policy, which yields consistent results in practice. Fig. 2 shows the empirical performance of controllers with different values of $\beta$ against time $k$.

From Fig. 2, one can observe that among our choices of $\beta$, only $\beta = 1/4$ drives the controller toward the optimal one. For $\beta = 0$, although the parameter estimates converge the fastest as discussed above, the performance of the resulting closed-loop system is even worse than the free system. This is because the exploration term $(k + 1)^{-\beta} \zeta_k$ in the control input does not decay, and the resulting controller is a noisy one. Meanwhile, for $\beta = 1/2$, diverging parameter estimates would lead to ill-performing controllers. The observations are consistent with the theoretical conclusion indicated by Theorem 3 that $\beta = 1/4$ corresponds to the optimal trade-off between exploration and exploitation.

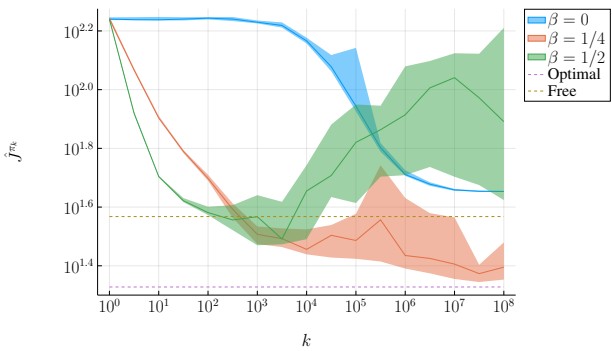

Figure 2: The empirical performance of controllers for different $\beta$ against time $k$. For each policy $\pi_k$, the empirical cost $\hat{J}^{\pi_k}$ is obtained by approximating the actual cost with random sampling from the closed-loop system. The solid lines are the median among the experiments, and the shades represent the interval between the first and third quantiles among the experiments. The purple and brown dashed lines represent the analytical optimal cost, and the analytical cost of the control-free system, respectively.

Finally, we compare our algorithm with the naive centainty equivalence algorithm, which applies $u_k = \hat{K}_k x_k + k^{-\beta}\zeta_k$ at each step. We invoke both algorithms with the "optimal" parameter $\beta = 1/4$, and perform 20 independent experiments for each algorithm. We consider the system parameter estimation error and the policy suboptimality gap for each algorithm, as we did in Fig. 1 and Fig. 2. The results of this comparison experiment are presented in Fig. 3. In a proportion of the experiments (actually 7 out of 20), certainty equivalence causes the estimation error and the policy suboptimality gap to diverge, while our algorithm ensures convergence in each experiment. Although existing works (e.g. Simchowitz & Foster (2020)) contain methods for decreasing the failure probability with a warm-up period for estimation, it should be noted that the failure probability is always nonzero as long as a linear feedback policy with learned gain is deployed. Even compared to the case where the certainty equivalence algorithm converges, our algorithm still delivers slightly lower estimation error and significantly better policy performance, which indicates that the conservativeness caused by our switching mechanism does not harm the long-term performance.

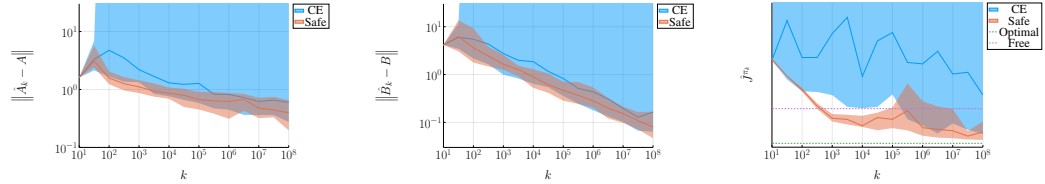

Figure 3: Comparison between our proposed algorithm (Safe) and the certainty equivalence algorithm (CE), in terms of estimation error and policy performance. In a proportion of the experiments, CE causes the estimation error and the policy suboptimality gap to diverge, while our algorithm ensures convergence in each experiment.

## 6 RELATED WORKS

The concept of dual control was initiated by Feldbaum (1960), and since then sustained research efforts have been devoted to learning to make decisions in unknown environments. Classical literature on adaptive control, e.g., Åström & Wittenmark (1973); Lai & Wei (1986); Guo & Chen (1988); Guo (1996), have addressed the dual control of linear systems described by the ARMAX model with the objective of tracking a reference signal. This objective do not consider the cost of exerting control input and is less involved than the LQR setting. For the rest of this section, we restrict our focus to the LQR of unknown linear systems described by the state-space model.

In the offline learning regime, system identification and controller synthesis based on the estimated system are performed in separate stages. The study of linear system identification has been developed for decades, with classical asymptotic guarantees summarized in Ljung (1999), and there has been a recent interest in non-asymptotic system identification based on a finite-size dataset. One can generate the dataset either by injecting random inputs to the system (Oymak & Ozay, 2019; Zheng & Li, 2020), or using active learning techniques to achieve optimal system-dependent finite-time rate (Wagenmaker & Jamieson, 2020). For a summary of recent results on non-asymptotic linear system identification, please refer to Zheng & Li (2020).

A natural step following system identification is to synthesize a controller based on the identification result and analyze the resulting closed-loop system's performance. The Coarse-ID control framework (Dean et al., 2019) adopts robust control techniques to derive controllers taking identification error into account, resulting in high-probability bounds on the control performance suboptimality gap. This framework has been applied to LQR (Dean et al., 2019), SISO system with output feedback (Boczar et al., 2018), and LQG (Zheng et al., 2020). As an alternative approach, the certainty equivalence framework synthesizes the controller by treating the identified system as the truth (Mania et al., 2019; Tsiamis et al., 2020), which is shown to achieve a faster convergence rate of control performance, at the cost of higher sensitivity to poorly estimated system parameters. At the intersection of the above two methods, an optimistic robust framework was proposed to achieve a combination of fast convergence rate and robustness (Umenberger & Schön, 2020). It should be pointed out the offline algorithms in the above works and the online setting we consider in the present work are not mutually exclusive but complementary to each other: online algorithms can continually refine the controller obtained by offline methods without interrupting normal system operation.

The problem of designing controllers that improve over time has been studied in the context of online LQR and regret analysis. The online LQR setting was initially proposed by Abbasi-Yadkori & Szepesvári (2011), and since then, Optimism in the Face of Uncertainty (OFU), Thompson Sampling (TS) and $\varepsilon$-greedy exploration have been applied to solve the online LQR problem. Abbasi-Yadkori & Szepesvári (2011) applied OFU principle to prove a theoretical $\tilde{\mathcal{O}}(\sqrt{T})$ regret upper bound, but their method is computationally intractable. Cohen et al. (2019) proposed a tractable algorithm for OFU based on semidefinite programming to realize $\tilde{\mathcal{O}}(\sqrt{T})$ regret. TS has been shown to achieve $\tilde{\mathcal{O}}(\sqrt{T})$ (frequentist) regret for scalar system (Abeille & Lazaric, 2018), and $\tilde{\mathcal{O}}(\sqrt{T})$ expected regret in a Beyesian setting (Ouyang et al., 2017). Recently, Faradonbeh et al. (2020a) and Mania et al. (2019) prove that randomized Certainty Equivalent (CE) control with $\varepsilon$-greedy exploration can also achieve $\tilde{\mathcal{O}}(\sqrt{T})$ regret, and Simchowitz & Foster (2020) prove that $\tilde{\mathcal{O}}(\sqrt{T})$ regret is indeed the fundamental limit for the general online LQR setting. Under further assumptions, e.g., partially known system parameters, logarithm regret bound may be achieved (Cassel et al., 2020). The aforementioned results and the fundamental limit hold either with high probability or in expectation, and lack almost sure guarantee. Faradonbeh et al. (2020b) prove that $\tilde{\mathcal{O}}(\sqrt{T})$ regret for both TS and CE and hold almost surely, but under the restrictive assumption that the closed-loop system remains stable when the policy is being employed. Our work attempts to fill in the gap by performing an almost sure analysis for online LQR without assumptions on the system other than the existence a known initial stabilizer.

# 7 CONCLUSION

In this paper, we propose a safe LQR dual control scheme based on a switching, which guarantees almost sure convergence to zero of both parameter inference error and suboptimality gap of control performance. The convergence rate of inference error is $\mathcal{O}(T^{-1/4+\epsilon})$, while that of the suboptimality gap is $\mathcal{O}(T^{-1/2+\epsilon})$, both of which match the known optimal rates proved in the non-asymptotic setting. For future works, we plan to extend the notion of regret to our almost-sure convergence setting, and formally establish the fundamental limits for this setting. It would also be interesting to investigate the exact convergence rates, especially dimension dependence, of our parameter inference scheme and the standard least-squares procedure.

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

# A  PROOF OF MAIN RESULTS

## A.1  PRELIMINARIES FOR PROOFS

### A.1.1  NOTATIONS

We will use the following notations throughout the proofs:

| Notation | Definition |
|---|---|
| $X^\top$ | transpose of matrix $X$ |
| $\|X\|, \|x\|$ | norm of matrix $X$ or vector $x$, 2-norm by default |
| $\|X\|_F$ | Frobenius norm of matrix $X$ |
| $\mathrm{tr}(X)$ | trace of matrix $X$ |
| $\rho(X)$ | spectral radius of matrix $X$ |
| $X \succ 0$ | matrix $X$ is positive definite |
| $X \prec Y$ | for matrix $X, Y, Y - X \succ 0$ |
| $\lambda_{\min}(X)$ | minimum eigenvalue of matrix $X$ |
| $\kappa(X)$ | ratio between maximum and minimum eigenvalues for matrix $X \succ 0$, i.e., $\kappa(X) = \|X\| / \lambda_{\min}(X)$ |
| $\mathrm{vec}(X)$ | vectorization of matrix $X$ |
| $X \otimes Y$ | Kronecker product of matrices $X$ and $Y$ |
| $\mathbf{1}_S$ | indicator function of set $S$ |
| $f(x) \sim \mathcal{O}(g(x))$ | there exists $M > 0$, such that $|f(k)| \le M \times g(k)$ for all $k \in \mathbb{N}$ |
| $f(x) \sim \tilde{\mathcal{O}}(g(x))$ | there exists $M > 0$ and $n \in \mathbb{N}$, such that $|f(k)| \le M \times g(k) \times \log(k)^n$ for all $k \in \mathbb{N}$ |

In addition, we define the following short-hand notation for characterizing the almost-sure asymptotic convergence or growth rate of random processes: for a random variable (vector, or matrix) sequence $\{x_k\}$, we denote that $x_k \sim \mathcal{C}(\alpha)$ if for all $\epsilon > 0$ and almost all realizations of randomness, we have $x_k \sim \mathcal{O}(k^{\alpha+\epsilon})$, i.e., $\lim_{k\to\infty} \|x_k\| / k^{\alpha+\epsilon} \overset{a.s.}{=} 0$. The following lemma establishes some basic properties of $\mathcal{C}(\alpha)$ functions:

**Lemma 2.** *Assume $x_k, y_k$ are random variable (vector, or matrix) sequences with proper dimensions, then the following properties hold:*

1. *If $x_k \sim \mathcal{C}(\alpha), y_k \sim \mathcal{C}(\beta)$, then $x_k + y_k \sim \mathcal{C}(\max(\alpha, \beta))$.*

2. *If $x_k \sim \mathcal{C}(\alpha), y_k \sim \mathcal{C}(\beta)$, then $x_k y_k \sim \mathcal{C}(\alpha + \beta)$.*

3. *If $f$ is a function differentiable at 0 and $x_k \sim C(\alpha), \alpha < 0$, then $f(x_k) - f(0) \sim C(\alpha)$.*

*Proof.* The first two claims follow trivially from the definition of $\mathcal{C}(\alpha)$. For the third claim, notice that for all realizations of randomness, $\lim_{k\to\infty} \|x_k\| = 0$. Therefore, Taylor expansion of $f(x)$ at $x = 0$ gives $f(x_k) - f(0) = Df(0)x_k + \mathcal{O}(\|x_k\|^2)$, where $Df$ is the Fréchet derivative of $f$. Dividing both sides by $k^{\alpha+\epsilon}$ leads to the conclusion. $\qquad\square$

### A.1.2  PROPERTIES OF A CLASS OF SWITCHED LINEAR SYSTEMS

We state and prove some properties of a class of switched linear systems that will be useful for the proof of all the main theorems of this paper.

Consider the dynamical system

$$x_{k+1} = A_k x_k + w_k, \tag{10}$$

where $x_k, w_k \in \mathbb{R}^n, x_0 \in \mathcal{N}(0, W_{-1}), w_k \sim \mathcal{N}(0, W_k), x_0, \{w_k \mid x_{0:k}\}$ are pairwise independent, and $A_k, W_k$ are functions of $x_{0:k}$ and $k$. We assume

$$W_k \succ 0, \|A_k\| \leq \mathscr{A}, \|W_k\| \leq \mathscr{W}, \quad \forall k. \tag{11}$$

We first prove a few miscellaneous lemmas that will be useful shortly:

**Lemma 3.** *Let $X \sim \chi^2(p)$, then $\mathbb{P}(X \geq M) \leq 2^{p/2} \exp(-M/4)$ for any $M > 0$.*

*Proof.* Applying the Chernoff bound, we have

$$\mathbb{P}(X \geq M) \leq \mathbb{E}\left[e^{tX}/e^{tM}\right] = (1 - 2t)^{-p/2} \exp(-tM)$$

for any $0 < t < 1/2$. Choosing $t = 1/4$ leads to the conclusion. $\qquad\square$

**Lemma 4.** *Let $w \in \mathbb{R}^p, w \sim \mathcal{N}(0, W), W \succ 0$, then $\mathbb{P}(\|w\| \geq M) \leq 2^{p/2} \exp\left(\frac{-M^2}{4\|W\|}\right)$.*

*Proof.* Let $v = W^{-1/2}w$, then $v \sim \mathcal{N}(0, I)$ and hence $\|v\|^2 \sim \chi^2(p)$. From $\|w\|^2 = w^\top w = v^\top W v \leq \|v\|^2 \|W\|$, we have $\mathbb{P}(\|w\| \geq M) = \mathbb{P}(\|w\|^2 \geq M^2) \leq \mathbb{P}(\|v\|^2 \geq M^2/\|W\|)$. Applying Lemma 3 leads to the conclusion. $\qquad\square$

**Lemma 5.** *Let $0 < x \leq 1/2$ and $1 < \alpha < 2$, then $\sum_{n=0}^{\infty} x^{\alpha^n} < 2x/(\alpha - 1)$.*

*Proof.* By $\alpha > 1$, we have $\alpha^n - 1 = (1 + \alpha - 1)^n - 1 > n(\alpha - 1)$. Therefore, in view of $0 < x < 1$, we have

$$\sum_{n=0}^{\infty} x^{\alpha^n} = x \sum_{n=0}^{\infty} x^{\alpha^n - 1} < x \sum_{n=0}^{\infty} x^{(\alpha-1)n} = \frac{x}{1 - x^{\alpha-1}}.$$

In view of the inequality $(1+t)^r \leq 1 + rt$ for $t \geq -1, 0 \leq r \leq 1$, we have $x^{\alpha-1} = (1+x-1)^{\alpha-1} \leq 1 + (\alpha - 1)(x - 1)$, and therefore,

$$\frac{x}{1 - x^{\alpha-1}} \leq \frac{x}{(\alpha - 1)(1 - x)} \leq \frac{2x}{\alpha - 1}$$

when $0 < x \leq 1/2$, and the conclusion follows. $\qquad\square$

Now we state several properties of the system described in (10)-(11):

**Lemma 6.** *For the system described in (10)-(11), assuming there exists $0 < \rho < 1$ and $P \succ 0$ such that $A_k^\top P A_k \prec \rho P$ for every $A_k$, then when $M \geq \sqrt{3\mathscr{W}\kappa(P)}/(1 - \rho^{1/4})$, there is*

$$\mathbb{P}(\|x_k\| \geq M) \leq \frac{2^{n/2+1}}{\rho^{-1/2} - 1} \exp\left(-\frac{(1 - \rho^{1/4})^2 M^2}{4\mathscr{W}\kappa(P)}\right), \quad \forall k.$$

*Proof.* From (10), we have

$$\begin{aligned}
x_k &= A_{k-1}x_{k-1} + w_{k-1} \\
&= A_{k-1}(A_{k-2}x_{k-2} + w_{k-2}) + w_{k-1} \\
&= \cdots \\
&= w_{k-1} + A_{k-1}w_{k-2} + \cdots + A_{k-1}A_{k-2}\cdots A_1 w_0 + A_{k-1}A_{k-2}\cdots A_0 x_0.
\end{aligned}$$

Pre-multiplying by $P^{1/2}$ on both sides and applying the triangle inequality, we have

$$\begin{aligned}
\left\|P^{1/2}x_k\right\| \leq \left\|P^{1/2}w_{k-1}\right\| + \left\|P^{1/2}A_{k-1}w_{k-2}\right\| + \cdots + \left\|P^{1/2}A_{k-1}\cdots A_1 w_0\right\| + \\
\left\|P^{1/2}A_{k-1}\cdots A_0 x_0\right\|. \tag{12}
\end{aligned}$$

From $A_k^\top P A_k \prec \rho P$, we have for any $w \in \mathbb{R}^n, k \in \mathbb{N}$,

$$w^\top A_k^\top P A_k w < \rho w^\top P w \Rightarrow \left\| P^{1/2} A_k w \right\| < \rho^{1/2} \left\| P^{1/2} w \right\|. \tag{13}$$

Applying (13) to (12) recursively, we have

$$\left\| P^{1/2} x_k \right\| \le \left\| P^{1/2} w_{k-1} \right\| + \rho^{1/2} \left\| P^{1/2} w_{k-2} \right\| + \cdots + \rho^{(k-1)/2} \left\| P^{1/2} w_0 \right\| + \rho^{k/2} \left\| P^{1/2} x_0 \right\|. \tag{14}$$

Meanwhile, we have

$$\{ \|x_k\| \ge M \} = \left\{ \left\| P^{-1/2} P^{1/2} x_k \right\| \ge M \right\}$$

$$\subseteq \left\{ \left\| P^{-1/2} \right\| \left\| P^{1/2} x_k \right\| \ge M \right\} = \left\{ \frac{1}{\sqrt{\lambda_{\min}(P)}} \left\| P^{1/2} x_k \right\| \ge M \right\}$$

$$= \left\{ \left\| P^{1/2} x_k \right\| \ge M \sqrt{\lambda_{\min}(P)} \right\}.$$

Therefore, for any $\sigma \in (\rho^{1/2}, 1)$, in view of (14), we have

$$\{ \|x_k\| \ge M \} \subseteq \left\{ \left\| P^{1/2} x_k \right\| \ge M \sqrt{\lambda_{\min}(P)} \right\}$$

$$\subseteq \left\{ \left\| P^{1/2} x_k \right\| \ge M \sqrt{\lambda_{\min}(P)} (1 - \sigma^{k+1}) \right\}$$

$$\subseteq \left\{ \left\| P^{1/2} w_{k-1} \right\| + \rho^{1/2} \left\| P^{1/2} w_{k-2} \right\| + \cdots + \rho^{(k-1)/2} \left\| P^{1/2} w_0 \right\| + \rho^{k/2} \left\| P^{1/2} x_0 \right\| \ge \right.$$
$$\left. M \sqrt{\lambda_{\min}(P)} (1 - \sigma^{k+1}) \right\}$$

$$\subseteq \left\{ \left\| P^{1/2} w_{k-1} \right\| \ge (1 - \sigma) M \sqrt{\lambda_{\min}(P)} \right\} \cup \left\{ \rho^{1/2} \left\| P^{1/2} w_{k-2} \right\| \ge \sigma(1 - \sigma) M \sqrt{\lambda_{\min}(P)} \right\} \cup$$

$$\cdots \cup \left\{ \rho^{(k-1)/2} \left\| P^{1/2} w_0 \right\| \ge \sigma^{k-1} (1 - \sigma) M \sqrt{\lambda_{\min}(P)} \right\} \cup$$

$$\left\{ \rho^{k/2} \left\| P^{1/2} x_0 \right\| \ge \sigma^k (1 - \sigma) M \sqrt{\lambda_{\min}(P)} \right\}.$$

Taking the union bound, we have

$$\mathbb{P}(\|x_k\| \ge M) \le \mathbb{P}\left( \left\| P^{1/2} w_{k-1} \right\| \ge (1 - \sigma) M \sqrt{\lambda_{\min}(P)} \right) +$$

$$\mathbb{P}\left( \left\| P^{1/2} w_{k-2} \right\| \ge \frac{\sigma}{\rho^{1/2}} (1 - \sigma) M \sqrt{\lambda_{\min}(P)} \right) +$$

$$\cdots +$$

$$\mathbb{P}\left( \left\| P^{1/2} w_0 \right\| \ge \left( \frac{\sigma}{\rho^{1/2}} \right)^{k-1} (1 - \sigma) M \sqrt{\lambda_{\min}(P)} \right) +$$

$$\mathbb{P}\left( \left\| P^{1/2} x_0 \right\| \ge \left( \frac{\sigma}{\rho^{1/2}} \right)^k (1 - \sigma) M \sqrt{\lambda_{\min}(P)} \right).$$

Since $P^{1/2} w_k \mid x_{0:k} \sim \mathcal{N}\left( 0, P^{1/2} W_k P^{1/2} \right)$, and $\|W_k\| \le \mathscr{W}$, applying Lemma 4, we have

$$\mathbb{P}\left( \left\| P^{1/2} w_{k-1} \right\| \ge (1 - \sigma) M \sqrt{\lambda_{\min}(P)} \middle| x_{0:k} \right) \le 2^{n/2} \exp\left( -\frac{(1 - \sigma)^2 M^2 \lambda_{\min}(P)}{4 \|W_k\| \|P\|} \right)$$

$$\le 2^{n/2} \exp\left( -\frac{(1 - \sigma)^2 M^2}{4 \mathscr{W} \kappa(P)} \right). \tag{15}$$

Noticing that the RHS of (15) does not depend on $x_{0:k}$, we have

$$\mathbb{P}\left( \left\| P^{1/2} w_{k-1} \right\| \ge (1 - \sigma) M \sqrt{\lambda_{\min}(P)} \right) \le 2^{n/2} \exp\left( -\frac{(1 - \sigma)^2 M^2}{4 \mathscr{W} \kappa(P)} \right).$$

Similarly, we have

$$\mathbb{P}\left(\left\|P^{1/2}w_{k-2}\right\| \geq \frac{\sigma}{\rho^{1/2}}(1-\sigma)M\sqrt{\lambda_{\min}(P)}\right) \leq \exp\left(-\frac{(1-\sigma)^2 M^2}{4\mathscr{W}\kappa(P)} \cdot \frac{\sigma^2}{\rho}\right),$$

$$\vdots$$

$$\mathbb{P}\left(\left\|P^{1/2}w_0\right\| \geq \left(\frac{\sigma}{\rho^{1/2}}\right)^{k-1}(1-\sigma)M\sqrt{\lambda_{\min}(P)}\right) \leq \exp\left(-\frac{(1-\sigma)^2 M^2}{4\mathscr{W}\kappa(P)} \cdot \left(\frac{\sigma^2}{\rho}\right)^{k-1}\right),$$

$$\mathbb{P}\left(\left\|P^{1/2}x_0\right\| \geq \left(\frac{\sigma}{\rho^{1/2}}\right)^{k}(1-\sigma)M\sqrt{\lambda_{\min}(P)}\right) \leq \exp\left(-\frac{(1-\sigma)^2 M^2}{4\mathscr{W}\kappa(P)} \cdot \left(\frac{\sigma^2}{\rho}\right)^{k}\right).$$

Choose $\sigma = \rho^{1/4}$, then $\sigma^2/\rho = \rho^{-1/2}$. Assuming w.l.o.g. that $\rho \in (1/4, 1)$, we have $\sigma^2/\rho = \rho^{-1/2} \in (1,2)$. When $M \geq \sqrt{3\mathscr{W}\kappa(P)}/(1-\rho^{1/4})$, there is

$$2^{n/2} \exp\left(-\frac{(1-\sigma)^2 M^2}{4\mathscr{W}\kappa(P)}\right) < \exp\left(-\frac{3}{4}\right) < \frac{1}{2}.$$

Applying Lemma 5 leads to the conclusion. $\qquad\square$

**Lemma 7.** *For the system described in* (10)-(11)*, assume there exists* $0 < \rho < 1, P \succ 0$ *and* $M > 0$ *such that* $A_k^\top P A_k \prec \rho P$ *as long as* $\|x_k\| \geq M$*. Let* $V_k = x_k^\top P x_k$*, then*

$$\mathbb{E}V_k < \frac{(M^2\mathscr{A}^2 + \mathscr{W})\|P\|}{1-\rho}, \quad \forall k.$$

*Proof.* We first derive a recursive bound on $\mathbb{E}V_k$: from (10), we have

$$V_{k+1} = x_{k+1}^\top P x_{k+1} = x_k^\top A_k^\top P A_k x_k + 2w_k^\top P A_k x_k + w_k^\top P w_k. \tag{16}$$

When $\|x_k\| < M$, we have $x_k^\top A_k^\top P A_k x_k \leq M^2\mathscr{A}^2\|P\|$. Otherwise, by $A_k^\top P A_k \prec \rho P$, we have $x_k^\top A_k^\top P A_k x_k < \rho x_k^\top P x_k = \rho V_k$. Therefore, in view of the fact that $M^2\mathscr{A}^2\|P\| \geq 0$ and $\rho V_k \geq 0$, we have

$$x_k^\top A_k^\top P A_k x_k \leq \max\{M^2\mathscr{A}^2\|P\|, \rho V_k \geq 0\} \leq M^2\mathscr{A}^2\|P\| + \rho V_k, \tag{17}$$

which holds for any $k$. Substituting (17) into (16), we have

$$V_{k+1} \leq \rho V_k + \eta_k + C, \tag{18}$$

where $\eta_k = 2w_k^\top P A_k x_k + w_k^\top P w_k, C = M^2\mathscr{A}^2\|P\|$. With such defined $\eta_k$, we have

$$\mathbb{E}\eta_k \leq \mathscr{W}\|P\|, \tag{19}$$

where we noticed that $\mathbb{E}[w_k^\top P A_k x_k] = \mathbb{E}[\mathbb{E}[w_k^\top P A_k x_k \mid x_{0:k}]] = \mathbb{E}[\mathbb{E}[w_k \mid x_{0:k}]^\top A_k x_k] = 0$, and that $\mathbb{E}[w_k^\top P w_k] = \mathbb{E}[\mathbb{E}[w_k^\top P w_k \mid x_{0:k}]] = \mathbb{E}[\mathrm{tr}(W_k P)] \leq \mathscr{W}\|P\|$. Taking the expectation on both sides of (18), we have

$$\mathbb{E}V_{k+1} \leq \rho\mathbb{E}V_k + (M^2\mathscr{A}^2 + \mathscr{W})\|P\|. \tag{20}$$

We can proceed using induction: we have

$$\mathbb{E}V_0 = \mathbb{E}[x_0^\top P x_0] = \mathbb{E}[\mathrm{tr}(W_{-1}P)] \leq \mathscr{W}\|P\| < \frac{(M^2\mathscr{A}^2 + \mathscr{W})\|P\|}{1-\rho}.$$

Assuming $\mathbb{E}V_k < \frac{(M^2\mathscr{A}^2+\mathscr{W})\|P\|}{1-\rho}$, it follows from (20) that $\mathbb{E}V_{k+1} < \frac{(M^2\mathscr{A}^2+\mathscr{W})\|P\|}{1-\rho}$, which concludes our proof. $\qquad\square$

**Lemma 8.** *For the system described in* (10)-(11)*, assume that there exists* $0 < \rho < 1, P \succ 0$ *and* $M > 0$ *such that* $A_k^\top P A_k \prec \rho P$ *as long as* $\|x_k\| \geq M$*, and that* $A_k$ *only depends on* $\|x_k\|$*. Let* $V_k = x_k^\top P x_k$*, then*

$$\mathbb{E}V_k^2 < \frac{(M^2\mathscr{A}^2 + \mathscr{W})\|P\|^2}{(1-\rho)(1-\rho^2)}\left[(1+\rho)(M^2\mathscr{A}^2 + \mathscr{W}) + 4\mathscr{A}^2\mathscr{W}\kappa(P)\right], \quad \forall k.$$

*Proof.* From the inequality (18) in the proof of Lemma 7, we have

$$\mathbb{E}V_{k+1}^2 \leq \rho^2 \mathbb{E}V_k^2 + \mathbb{E}\eta_k^2 + C^2 + 2\rho\mathbb{E}[V_k\eta_k] + 2\rho C\mathbb{E}V_k + 2C\mathbb{E}\eta_k, \tag{21}$$

where $\eta_k = 2w_k^\top P A_k x_k + w_k^\top P w_k, C = M^2\mathscr{A}^2\|P\|$. Let us bound $\mathbb{E}\eta_k, \mathbb{E}V_k, \mathbb{E}\eta_k^2, \mathbb{E}[V_k\eta_k]$ which appear in the RHS of (21) respectively:

- $\mathbb{E}\eta_k \leq \mathscr{W}\|P\|$ from (19) in the proof of Lemma 7.

- $\mathbb{E}V_k < (C + \mathscr{W}\|P\|)/(1 - \rho)$ according to the conclusion of Lemma 7.

- $\mathbb{E}\eta_k^2 = 4\mathbb{E}[x_k^\top A_k^\top P w_k w_k^\top P A_k x_k] + 4\mathbb{E}[w_k^\top P A_k x_k w_k^\top P w_k] + \mathbb{E}[w_k^\top P w_k w_k^\top P w_k] \leq 4\mathbb{E}\|x_k\|^2 \mathscr{A}^2\mathscr{W}\|P\|^2 + \mathscr{W}^2\|P\|^2$, where we noticed $\mathbb{E}[w_k^\top P A_k x_k w_k^\top P w_k] = \mathbb{E}[\mathbb{E}[w_k^\top P A_k x_k w_k^\top P w_k \mid x_k]] = \mathrm{tr}\left\{\mathbb{E}[A_k x_k \mathbb{E}[w_k^\top P w_k w_k^\top P \mid x_k]]\right\} = 0$ due to symmetry. From $V_k = x_k^\top P x_k \geq \lambda_{\min}(P)\|x_k\|^2$, we can further deduce $\mathbb{E}\eta_k^2 \leq 4\mathscr{A}^2\mathscr{W}\|P\|\kappa(P)\mathbb{E}V_k + \mathscr{W}^2\|P\|^2$.

- $\mathbb{E}[V_k\eta_k] = 2\mathbb{E}[w_k^\top P A_k V_k] + \mathbb{E}[V_k w_k^\top P w_k] = 2\mathbb{E}[\mathbb{E}[w_k^\top P A_k x_k V_k \mid x_k]] + \mathbb{E}[\mathbb{E}[v_k w_k^\top P w_k \mid x_k]] = 2\mathbb{E}[\mathbb{E}[w_k \mid x_k]^\top P A_k x_k V_k] + \mathbb{E}[V_k\mathbb{E}[w_k^\top P w_k \mid x_k]] \leq \mathscr{W}\|P\|\mathbb{E}V_k$.

Substituting the above terms into the RHS of (21), we have

$$\mathbb{E}V_{k+1}^2 < \rho^2\mathbb{E}V_k^2 + \left(4\mathscr{A}^2\mathscr{W}\|P\|\kappa(P) + 2\rho(C + \mathscr{W}\|P\|)\right)\frac{C + \mathscr{W}\|P\|}{1 - \rho} +$$
$$C^2 + 2C\mathscr{W}\|P\| + \mathscr{W}^2\|P\|^2$$
$$= \rho^2\mathbb{E}V_k^2 + \frac{C + \mathscr{W}\|P\|}{1 - \rho}\left[(1 + \rho)(C + \mathscr{W}\|P\|) + 4\mathscr{A}^2\mathscr{W}\|P\|\kappa(P)\right]. \tag{22}$$

We can then proceed using induction: we have

$$\mathbb{E}V_0^2 \leq \mathscr{W}^2\|P\|^2 = (\mathscr{W}\|P\|)(\mathscr{W}\|P\|)$$
$$< \frac{C + \mathscr{W}\|P\|}{1 - \rho}(C + \mathscr{W}\|P\|)$$
$$< \frac{C + \mathscr{W}\|P\|}{(1 - \rho)(1 - \rho^2)}\left[(1 + \rho)(C + \mathscr{W}\|P\|) + 4\mathscr{A}^2\mathscr{W}\|P\|\kappa(P)\right].$$

Assuming $\mathbb{E}V_k < \frac{C+\mathscr{W}\|P\|}{(1-\rho)(1-\rho^2)}\left[(1+\rho)(C+\mathscr{W}\|P\|) + 4\mathscr{A}^2\mathscr{W}\|P\|\kappa(P)\right]$, we also have $\mathbb{E}V_{k+1} < \frac{C+\mathscr{W}\|P\|}{(1-\rho)(1-\rho^2)}\left[(1+\rho)(C+\mathscr{W}\|P\|) + 4\mathscr{A}^2\mathscr{W}\|P\|\kappa(P)\right]$ by (22). Therefore, we have

$$\mathbb{E}V_k < \frac{C + \mathscr{W}\|P\|}{(1 - \rho)(1 - \rho^2)}\left[(1 + \rho)(C + \mathscr{W}\|P\|) + 4\mathscr{A}^2\mathscr{W}\|P\|\kappa(P)\right]$$
$$= \frac{(M^2\mathscr{A}^2 + \mathscr{W})\|P\|^2}{(1 - \rho)(1 - \rho^2)}\left[(1 + \rho)(M^2\mathscr{A}^2 + \mathscr{W}) + 4\mathscr{A}^2\mathscr{W}\kappa(P)\right]$$

for any $k$. $\qquad\square$

A linear dynamical system (1) driven by our safe switching policy can be viewed as an instance of the system described in (10)-(11). Formally, we consider the following system:

$$x_{k+1} = Ax_k + Bu_k + w_k, \tag{23}$$

We consider the following two classes of policies:

- Linear feedback policy: $u_k = \pi^K(x_k) = Kx_k$.
- Safe switching policy: $(u_k, \xi_{k+1}) = \bar{\pi}^{K,M,t}(x_k, \xi_k)$, where $M > 0$ is the switching threshold and $t \in \mathbb{N}^*$ is the "non-action" duration, and $\bar{\pi}^{K,M,t}$ determines $u_k, \xi_{k+1}$ as follows:
    - If $\xi_k > 0$, then $u_k = 0, \xi_{k+1} = \xi_k - 1$;

- If $\xi_k = 0, \max\{\|K\|, \|x_k\|\} \geq M$, then $u_k = 0, \xi_{k+1} = t - 1$;
- If $\xi_k = 0, \max\{\|K\|, \|x_k\|\} < M$, then $u_k = Kx_k, \xi_{k+1} = 0$.

Notably, the linear feedback policy can be viewed as a special case of the safe switching policy where the switching threshold is infinity, i.e., $\pi^K = \bar{\pi}^{K,+\infty,1}$.

For the simplicity of expressions, when a safe switching policy $\bar{\pi}^{K_k,M_k,t_k}$ is applied at each step $k$, let us define the following notations:

$$i(0) = 0, i(k+1) = \begin{cases} i(k) + 1 & u_{i(k)} \neq 0 \\ i(k) + t_{i(k)} & u_{i(k)} = 0 \end{cases}, \tilde{x}_k = x_{i(k)}. \tag{24}$$

In plain words, $\{\tilde{x}_k\} = \{x_{i(k)}\}$ is the subsequence of $\{x_k\}$ where "non-action" steps are skipped. In addition, let

$$\mathcal{W} = \left\| \lim_{t \to \infty} W + AWA^\top + \cdots + A^t W (A^t)^\top \right\|, \tag{25}$$

which is a finite value since $A$ is stable.

**Lemma 9.** *For the system described in (23)-(25), assume a safe switching policy $\bar{\pi}^{K_k,M_k,t_k}$ is applied at each step $k$, $M_k \leq M$ for any $k$, and there exists $0 < \rho < 1$ and $P \succ 0$ such that $A^\top P A \prec \rho P$, then*

$$\mathbb{E} \|x_k\|^4 < 8 \left[ \mathcal{Q}(M, \rho, P) + \mathcal{W}^2 \kappa(P)^2 \right],$$

*where*

$$\mathcal{Q}(M, \rho, P) = \frac{(M^2 \mathcal{A}^2 + \mathcal{W}) \|P\|^2}{(1 - \rho)(1 - \rho^2)} \left[ (1 + \rho)(M^2 \mathcal{A}^2 + \mathcal{W}) + 4\mathcal{A}^2 \mathcal{W} \kappa(P) \right],$$

$$\mathcal{A} = \|A\| + \|B\| M.$$

*Proof.* According to (23) and (24), the state trajectory $\{\tilde{x}_k\}$ evolves subject to the dynamics

$$\tilde{x}_{k+1} = A_k \tilde{x}_k + \tilde{w}_k, \tag{26}$$

where

$$A_k = \begin{cases} A & \max\{\|K_k\|, \|\tilde{x}_k\|\} \geq M \\ A + BK_k & \text{otherwise} \end{cases},$$

$$\tilde{w}_k \mid \tilde{x}_k \sim \mathcal{N}(0, W_k), W_k = \begin{cases} W & \max\{\|K_k\|, \|\tilde{x}_k\|\} \geq M \\ W + AWA^\top + \cdots + A^{t_k} W (A^{t_k})^\top & \text{otherwise} \end{cases}.$$

Let $V_k = x_k^\top P x_k$ and $\tilde{V}_k = \tilde{x}_k^\top P \tilde{x}_k$. Since $A^\top P A \prec \rho P$, the system (26) satisfies the assumption of Lemma 8, and $\|W_k\| \leq \mathcal{W}, \|A_k\| \leq \|A\| + \|B\| M = \mathcal{A}$. Therefore, and therefore,

$$\mathbb{E}\tilde{V}_k^2 < \frac{(M^2 \mathcal{A}^2 + \mathcal{W}) \|P\|^2}{(1 - \rho)(1 - \rho^2)} \left[ (1 + \rho)(M^2 \mathcal{A}^2 + \mathcal{W}) + 4\mathcal{A}^2 \mathcal{W} \kappa(P) \right] = \mathcal{Q}(M, \rho, P) \lambda_{\min}(P)^2. \tag{27}$$

Next, to establish the relationship between $\mathbb{E}\tilde{V}_k^2$ and $\mathbb{E}V_k^2$, notice that

$$x_k = A^{k-i(j)} \tilde{x}_j + \tilde{w}_k, \tag{28}$$

where $j = \sup\{s \in \mathbb{N} \mid i(s) \leq k\}$, i.e., $\tilde{x}_j$ is the last state in $\{\tilde{x}_k\}$ that physically occurs no later than $x_k$, and $\tilde{w}_k = \sum_{l=0}^{k-i(j)-2} A^{k-i(j)-1-l} w_{i(j)+l}$, Since $A$ is stable, we have $\|\mathbb{E}[\tilde{w}_k \tilde{w}_k^\top]\| \leq \mathcal{W}$. Pre-multiplying both sides of (28) with $P^{1/2}$ and applying the triangle inequality, we have

$$\left\| P^{1/2} x_k \right\| \leq \left\| P^{1/2} A^{k-i(j)} \tilde{x}_j \right\| + \left\| P^{1/2} \tilde{w}_k \right\|.$$

Therefore, applying the power means inequality $(\frac{a+b}{2})^4 \le \frac{a^4+b^4}{2}$, we have

$$
\begin{aligned}
V_k^2 = \left\| P^{1/2} x_k \right\|^4 &\le 8 \left( \left\| P^{1/2} A^{k-i(j)} \tilde{x}_j \right\|^4 + \left\| P^{1/2} \tilde{w}_k \right\|^4 \right) \\
&= 8 \left( \left( \tilde{x}_j^\top \left( A^{k-i(j)} \right)^\top P A^{k-i(j)} \tilde{x}_j \right)^2 + \left( \tilde{w}_k^\top P \tilde{w}_k \right)^2 \right) \\
&\le 8 \left( \rho^{2(k-i(j))} \tilde{V}_j^2 + \left( \tilde{w}_k^\top P \tilde{w}_k \right)^2 \right).
\end{aligned}
$$

Taking the expectation on both sides of the above inequality and applying (27), we get

$$
\mathbb{E} V_k^2 \le 8 \left[ \mathcal{Q}(M, \rho, P) \lambda_{\min}(P)^2 + \mathscr{W}^2 \|P\|^2 \right].
$$

Finally, using the fact that $V_k = x_k^\top P x_k \ge \lambda_{\min}(P) \|x_k\|^2$, we have

$$
\mathbb{E} \|x_k\|^4 \le \frac{\mathbb{E} V_k^2}{\lambda_{\min}(P)^2} \le 8 \left[ \mathcal{Q}(M, \rho, P) + \mathscr{W}^2 \kappa(P)^2 \right],
$$

which concludes our proof. $\qquad\square$

**Lemma 10.** *For the system described in (23)-(25), assume $K$ is a stabilizing linear feedback (i.e., $\rho(A + BK) < 1$) gain with $\|K\| \le M$, and there exists $0 < \rho < 1$ and $P \succ 0$ such that $A^\top P A \prec \rho P, (A + BK)^\top P (A + BK) \prec \rho P$. Let $J$ be the cost (defined in (3)) associated with the linear feedback policy $\pi^K$, and $J^{M,t}$ be the cost associated with the safe switching policy $\bar{\pi}^{K,M,t}$. Then*

$$
J^{M,t} - J \le 2\mathcal{C}_1(\rho, P, K) \mathcal{G}(M, t, \rho, P, K) + \mathcal{G}(M, t, \rho, P, K)^2,
$$

*where*

$$
\mathcal{C}_1(\rho, P, K) = \left( \frac{\mathscr{W} \kappa(P) \|Q + K^\top R K\|}{1 - \rho} \right)^{1/2},
$$

$$
\mathcal{G}(M, t, \rho, P, K) = \mathcal{C}_2(\rho, K) t \left[ \mathcal{Q}(M, \rho, P) + \mathscr{W}^2 \kappa(P)^2 \right]^{1/4} \mathcal{E}(M, \rho, P),
$$

$$
\mathcal{C}_2(\rho, K) = \|Q + K^\top R K\| \|BK\| \cdot \sum_{s=0}^\infty \|(A + BK)^s\| \cdot \frac{2^{n/2 + 7/4}}{\rho^{-1/2} - 1},
$$

$$
(\mathcal{C}_2(\rho, K) < \infty \text{ because } \rho(A + BK) < 1),
$$

$$
\mathcal{Q}(M, \rho, P) = \frac{(M^2 \mathscr{A}^2 + \mathscr{W}) \|P\|^2}{(1 - \rho)(1 - \rho^2)} \left[ (1 + \rho)(M^2 \mathscr{A}^2 + \mathscr{W}) + 4\mathscr{A}^2 \mathscr{W} \kappa(P) \right],
$$

$$
\mathcal{E}(M, \rho, P) = \exp \left( -\frac{(1 - \rho^{1/4})^2 M^2}{4 \mathscr{W} \kappa(P)} \right).
$$

*In particular, if the system specification $A, B, Q, R, \mathscr{W}$, the stabilizing linear feedback gain $K$, and the parameters $\rho, P$ are all fixed, we have*

$$
J^{M,t} - J \sim \mathcal{O} \left( tM \exp(-cM^2) \right)
$$

*as $M \to \infty, t \to \infty, tM \exp(-cM^2) \to 0$, where $c = \frac{(1-\rho^{1/4})^2}{4\mathscr{W}\kappa(P)}$.*

*Proof.* Let $\{x_k\}, \{u_k\}$ denote the state and input trajectories driven by $\pi^K$, and $\{\bar{x}_k\}, \{\bar{u}_k\}$ denote the state and input trajectories driven by $\bar{\pi}^{K,M,t}$. Then from $\pi^K(x) = Kx$ and $\bar{\pi}^{K,M,t}(x) \in \{Kx, 0\}$, we have

$$
J = \lim_{T \to \infty} \mathbb{E} \left[ \frac{1}{T} \sum_{k=0}^{T-1} x_k^\top (Q + K^\top R K) x_k \right] = \lim_{T \to \infty} \frac{1}{T} \sum_{k=0}^{T-1} \mathbb{E} \left[ x_k^\top (Q + K^\top R K) x_k \right],
$$

$$
J^{M,t} \le \limsup_{T \to \infty} \mathbb{E} \left[ \frac{1}{T} \sum_{k=0}^{T-1} \bar{x}_k^\top (Q + K^\top R K) \bar{x}_k \right] = \limsup_{T \to \infty} \frac{1}{T} \sum_{k=0}^{T-1} \mathbb{E} \left[ \bar{x}_k^\top (Q + K^\top R K) \bar{x}_k \right].
$$

Therefore, we only need to prove

$$\mathbb{E}\left[\bar{x}_k^\top(Q + K^\top RK)\bar{x}_k\right] - \mathbb{E}\left[x_k^\top(Q + K^\top RK)x_k\right] \leq$$
$$2\mathcal{C}_1(\rho, P, K)\mathcal{G}(M, t, \rho, P, K) + \mathcal{G}(M, t, \rho, P, K)^2$$

for every $k$. Let $\tilde{Q} = Q + K^\top RK$. Since $Q \succ 0$ and $R \succ 0$, we have $\tilde{Q} \succ 0$, and therefore $\|x\|_{\tilde{Q}} \triangleq \sqrt{x^\top\tilde{Q}x}$ defines a norm. In the follows, we bound $\mathbb{E}\|\bar{x}_k\|_{\tilde{Q}}^2 - \mathbb{E}\|x_k\|_{\tilde{Q}}^2$.

We notice that

$$\begin{aligned}
\bar{x}_k &= (A + BK)\bar{x}_{k-1} + w_{k-1} - BK\bar{x}_{k-1}\mathbf{1}_{\{\bar{u}_{k-1}=0\}} \\
&= (A + BK)\left[(A + BK)\bar{x}_{k-2} + w_{k-2} - BK\bar{x}_{k-2}\mathbf{1}_{\{\bar{u}_{k-2}=0\}}\right] + w_{k-1} - \\
&\qquad BK\bar{x}_{k-1}\mathbf{1}_{\{\bar{u}_{k-1}=0\}} \\
&= \cdots \\
&= (A + BK)^k x_0 + \sum_{s=0}^{k-1}(A + BK)^s w_{k-s-1} - \sum_{s=0}^{k-1}(A + BK)^s BK\bar{x}_{k-s-1}\mathbf{1}_{\{\bar{u}_{k-s-1}=0\}} \\
&= x_k - \sum_{s=0}^{k-1}(A + BK)^s BK\bar{x}_{k-s-1}\mathbf{1}_{\{\bar{u}_{k-s-1}\}}.
\end{aligned}$$

Hence,

$$\|\bar{x}_k\|_{\tilde{Q}} \leq \|x_k\|_{\tilde{Q}} + \sum_{s=0}^{k-1}\left\|(A + BK)^s BK\bar{x}_{k-s-1}\mathbf{1}_{\{\bar{u}_{k-s-1}\}}\right\|_{\tilde{Q}}$$

$$\leq \|x_k\|_{\tilde{Q}} + \left\|\tilde{Q}\right\|\|BK\|\sum_{s=0}^{k-1}\|(A + BK)^s\|\left\|\bar{x}_{k-s-1}\mathbf{1}_{\{\bar{u}_{k-s-1}\}}\right\|.$$

From the fact that $\mathbb{E}(X_1 + \cdots + X_n)^2 \leq \left(\sqrt{\mathbb{E}X_1^2} + \cdots + \sqrt{\mathbb{E}X_n^2}\right)^2$, where $X_1, \ldots, X_n$ are arbitrary random variables with bounded second-order moments, we have

$$\mathbb{E}\|\bar{x}_k\|_{\tilde{Q}}^2 \leq \left(\sqrt{\mathbb{E}\|x_k\|_{\tilde{Q}}^2} + \left\|\tilde{Q}\right\|\|BK\|\sum_{s=0}^{k-1}\|(A + BK)^s\|\sqrt{\mathbb{E}\left[\left\|\bar{x}_{k-s-1}\right\|^2\mathbf{1}_{\{\bar{u}_{k-s-1}=0\}}\right]}\right)^2.$$

By Cauchy-Schwarz inequality, we have

$$\mathbb{E}\|\bar{x}_k\|_{\tilde{Q}}^2 \leq \left(\sqrt{\mathbb{E}\|x_k\|_{\tilde{Q}}^2} + \left\|\tilde{Q}\right\|\|BK\|\sum_{s=0}^{k-1}\|(A + BK)^s\|\left(\mathbb{E}\|\bar{x}_{k-s-1}\|^4\right)^{\frac{1}{4}}\mathbb{P}\left(\bar{u}_{k-s-1} = 0\right)\right)^2.$$

By Lemma 9, we have $\mathbb{E}\|\bar{x}_{k-s-1}\|^4 \leq 8[\mathcal{Q}(M, \rho, P) + \mathscr{W}^2\kappa(P)^2]$ for any $s$, and hence,

$$\begin{aligned}
\mathbb{E}\|\bar{x}_k\|_{\tilde{Q}}^2 \leq \Big(&\sqrt{\mathbb{E}\|x_k\|_{\tilde{Q}}^2} + \left\|\tilde{Q}\right\|\|BK\|2^{3/4}[\mathcal{Q}(M, \rho, P) + \mathscr{W}^2\kappa(P)^2]^{1/4} \\
&\sum_{s=0}^{k-1}\|(A + BK)^s\|\mathbb{P}\left(\bar{u}_{k-s-1} = 0\right)\Big)^2.
\end{aligned} \tag{29}$$

According to the update rule of $\xi_k$ in $\bar{\pi}^{K,M,t}$, we have

$$\{\bar{u}_k = 0\} \subseteq \bigcup_{\tau=0}^{t-1}\{\|\bar{x}_{k-t}\| \geq M\}.$$

Taking the union bound, we have

$$\mathbb{P}\left(\bar{u}_k = 0\right) \leq \sum_{\tau=0}^{t-1}\mathbb{P}\left(\|\bar{x}_{k-t}\| \geq M\right).$$

By Lemma 6, we have

$$\mathbb{P}\left(\|\bar{x}_{k-t}\| \geq M\right) \leq \frac{2^{n/2+1}}{\rho^{-1/2}-1} \mathcal{E}(M, \rho, P)$$

for every $k$ and $t$, and hence

$$\mathbb{P}\left(\bar{u}_k = 0\right) \leq t \frac{2^{n/2+1}}{\rho^{-1/2}-1} \mathcal{E}(M, \rho, P) \tag{30}$$

for every $k$. Substituting (30) into (29), we get

$$\mathbb{E}\|\bar{x}_k\|_{\tilde{Q}}^2 \leq \left( \sqrt{\mathbb{E}\|x_k\|_{\tilde{Q}}^2} + \left\|\tilde{Q}\right\| \|BK\| \frac{2^{n/2+7/4}}{\rho^{-1/2}-1} [\mathcal{Q}(M, \rho, P) + \mathscr{W}^2 \kappa(P)^2]^{1/4} \right.$$
$$\left. \sum_{s=0}^{k-1} \|(A+BK)^s\| \mathcal{E}(M, \rho, P) \right)^2$$

$$\leq \left( \sqrt{\mathbb{E}\|x_k\|_{\tilde{Q}}^2} + \mathcal{C}_2(\rho, K)[\mathcal{Q}(M, \rho, P) + \mathscr{W}^2 \kappa(P)^2]^{1/4} \mathcal{E}(M, \rho, P) \right)^2$$

$$= \left( \sqrt{\mathbb{E}\|x_k\|_{\tilde{Q}}^2} + \mathcal{G}(M, t, \rho, P, K) \right)^2$$

$$= \mathbb{E}\|x_k\|_{\tilde{Q}}^2 + 2\sqrt{\mathbb{E}\|x_k\|_{\tilde{Q}}^2} \mathcal{G}(M, t, \rho, P, K) + \mathcal{G}(M, t, \rho, P, K)^2.$$

Applying Lemma 7 with $M = 0$, we have

$$\mathbb{E}[x_k^\top P x_k] \leq \frac{\mathscr{W}\|P\|}{1-\rho},$$

and hence

$$\mathbb{E}\|x_k\|_{\tilde{Q}}^2 \leq \left\|\tilde{Q}\right\| \|x_k\|^2 \leq \frac{\mathscr{W}\kappa(P)\|Q\|}{1-\rho} = \mathcal{C}_1(\rho, P, K)^2.$$

Therefore,

$$\mathbb{E}\|\bar{x}_k\|_{\tilde{Q}}^2 - \mathbb{E}\|x_k\|_{\tilde{Q}}^2 \leq 2\mathcal{C}_1(\rho, P, K)\mathcal{G}(M, t, \rho, P, K) + \mathcal{G}(M, t, \rho, P, K)^2,$$

and hence

$$J^{M,t} - J \leq 2\mathcal{C}_1(\rho, P, K)\mathcal{G}(M, t, \rho, P, K) + \mathcal{G}(M, t, \rho, P, K)^2,$$

which concludes our proof. $\qquad\square$

### A.1.3 A LAW OF LARGE NUMBERS FOR MARTINGALES

Let $\{\mathcal{F}_k\}$ be a filtration of $\sigma$-algebras and $\{S_k\}$ be a matrix-valued stochastic process adapted to $\{\mathcal{F}_k\}$, we call $\{S_k\}$ a matrix-valued martingale (with respect to the filtration $\{\mathcal{F}_k\}$) if $\mathbb{E}[S_{k+1} \mid \mathcal{F}_k] = S_k$ holds for all $k$. Now we can state a law of large numbers for matrix-valued martingales:

**Lemma 11.** *(Liu et al., 2020, Lemma 4) If $S_k = \Phi_0 + \Phi_1 + \cdots + \Phi_k$ is a matrix-valued martingale such that*

$$\mathbb{E}\|\Phi_k\|^2 \sim \mathcal{C}(\beta),$$

*where $0 \leq \beta < 1$, then $S_k/k$ converges to 0 almost surely. Furthermore,*

$$\frac{S_k}{k} \sim \mathcal{C}\left(\frac{\beta-1}{2}\right).$$

### A.1.4 A PERTURBATION ANALYSIS OF ALGEBRAIC RICCATI EQUATION

An interesting property of LQR is that near the optimal controller (which corresponds to the solution of the discrete algebraic Riccati equation), the perturbation of the discrete Lyapunov equation solution is quadratic in the perturbation of controller gain. Basically, this results in the convergence speed of the certainty equivalent control performance being twice that of the estimation error. Similar results have been reported in (Mania et al., 2019; Simchowitz & Foster, 2020), and here we present a simplified version that would suffice for our purpose.

**Lemma 12.** *Consider the discrete Lyapunov equation*

$$A^\top X A - X + Q = 0,$$

*where $Q \succ 0$ and $\rho(A) < 1$, then the unique positive definite solution $X$ satisfies*

$$\|X\|_F \leq \left\| \left( I - A^\top \otimes A^\top \right)^{-1} \right\|_2 \|Q\|_F.$$

*Proof.* Vectorizing the Lyapunov equation, we get

$$\mathrm{vec}(X) = \left( I - A^\top \otimes A^\top \right)^{-1} \mathrm{vec}(Q),$$

and hence

$$\|X\|_F = \|\mathrm{vec}(X)\|_2 \leq \left\| \left( I - A^\top \otimes A^\top \right)^{-1} \right\|_2 \|\mathrm{vec}(Q)\|_2 = \left\| \left( I - A^\top \otimes A^\top \right)^{-1} \right\|_2 \|Q\|_F.$$

$\square$

**Lemma 13.** *Let $P$ be the solution to the discrete algebraic Riccati equation*

$$P = Q + A^\top P A - A^\top P B \left( R + B^\top P B \right)^{-1} B^\top P A,$$

*and let $K$ be defined as*

$$K = - \left( R + B^\top P B \right)^{-1} B^\top P A.$$

*Assume $\hat{K}$ is such that $\tilde{A} = A + B\hat{K}$ is stable, and $\hat{P}$ satisfies the discrete Lyapunov equation*

$$\hat{P} = Q + \hat{K}^\top R \hat{K} + \tilde{A}^\top \hat{P} \tilde{A}. \tag{31}$$

*Let $\Delta P = \hat{P} - P, \Delta K = \hat{K} - K$, then*

$$\|\Delta P\|_F \leq \left\| \left( I - \tilde{A}^\top \otimes \tilde{A}^\top \right)^{-1} \right\|_2 \|R\|_F \|\Delta K\|_F^2.$$

*Proof.* It can be shown $P$ satisfies the discrete Lyapunov equation

$$P = Q + K^\top R K + (A + BK)^\top P (A + BK). \tag{32}$$

Meanwhile, substituting $\Delta P = \hat{P} - P, \Delta K = \hat{K} - K$ into (31), we get

$$P + \Delta P$$
$$= Q + (K + \Delta K)^\top R (K + \Delta K) + (A + B (K + \Delta K))^\top (P + \Delta P) (A + B (K + \Delta K))$$
$$= Q + K^\top R K + \Delta K^\top R \Delta K + (A + BK)^\top P (A + BK) + \tilde{A}^\top \Delta P \tilde{A}, \tag{33}$$

where for the second equality we used the fact

$$\left( R + B^\top P B \right) K + B^\top P A = 0.$$

By taking the difference between (33) and (32), we can see $\Delta P$ also satisfies a discrete Lyapunov equation:

$$\Delta P = \Delta K^\top R \Delta K + \tilde{A}^\top \Delta P \tilde{A}.$$

Applying Lemma 12, we obtain

$$\|\Delta P\|_F \leq \left\| \left( I - \tilde{A}^\top \otimes \tilde{A}^\top \right)^{-1} \right\|_2 \left\| \Delta K^\top R \Delta K \right\|_F$$

$$\leq \left\| \left( I - \tilde{A}^\top \otimes \tilde{A}^\top \right)^{-1} \right\|_2 \|R\|_F \|\Delta K\|_F^2,$$

which concludes our proof.

$\square$

## A.2 Proof of Theorem 1

*Proof.* According to our definition of bounded-cost safety, we only need to prove $J^{\pi_k} < +\infty$ for every $k$, where $J^{\pi_k}$ is defined in (3).

Let $\{x_i\}, \{u_i\}$ denote the state and input trajectories driven by $\pi_k$. According to Algorithm 2, we have $u_i \in \{\hat{K}_k x_i, 0\}$, and therefore,

$$J^{\pi_k} \leq \limsup_{T \to \infty} \mathbb{E}\left[\frac{1}{T}\sum_{i=0}^{T-1} x_i^\top \left(Q + \hat{K}_k^\top R\hat{K}_k\right)x_i\right] = \limsup_{T \to \infty}\frac{1}{T}\sum_{i=0}^{T-1}\mathbb{E}\left[x_i^\top\left(Q + \hat{K}_k^\top R\hat{K}_k\right)x_i\right].$$

Notice that under $\pi_k$, there is $x_{i+1} = Ax_i + w_i$ as long as $\|x_i\| \geq \log k$, and since $A$ is stable, there exists $0 < \rho < 1$ and $P \succ 0$ such that $A^\top PA \prec \rho P$. By Lemma 7, we have

$$\mathbb{E}\left[x_i^\top Px_i\right] < \frac{((\log k)^2 \mathscr{A}^2 + \|W\|)\|P\|}{1-\rho},$$

where $\mathscr{A} = \max\{\|A\|, \|A + B\hat{K}_k\|\}$. Therefore, by $x_i^\top Px_i \geq \lambda_{\min}(P)\|x_i\|^2$ and $x_i^\top\left(Q + \hat{K}_k^\top R\hat{K}_k\right)x_i \leq \|Q + \hat{K}_k^\top R\hat{K}_k\|\|x_i\|^2$, we have

$$\mathbb{E}\left[x_i^\top\left(Q + \hat{K}_k^\top R\hat{K}_k\right)x_i\right] < \frac{((\log k)^2\mathscr{A}^2 + \|W\|)\kappa(P)\|Q + \hat{K}_k^\top R\hat{K}_k\|}{1-\rho}.$$

This implies

$$J^{\pi_k} \leq \frac{((\log k)^2\mathscr{A}^2 + \|W\|)\kappa(P)\|Q + \hat{K}_k^\top R\hat{K}_k\|}{1-\rho} < +\infty,$$

which concludes our proof. $\square$

## A.3 Proof of Theorem 2

*Proof.* We shall assume throughout the proof that $\{\mathcal{F}_k\}$ is the $\sigma$-algebra generated by the random variables $\{x_0, w_0, \ldots, w_k, \zeta_0, \ldots, \zeta_k\}$.

We first cast the system (1) into a static form by writing $x_k$ as

$$x_k = A^k x_0 + \sum_{t=0}^{k-1} A^t Bu_{k-t-1} + \sum_{t=0}^{k-1} A^t w_{k-t-1}$$

$$= A^k x_0 + \sum_{t=0}^{k-1} H_t\left[\tilde{u}_{k-t-1} + (k-t)^{-\beta}\zeta_{k-t-1}\right] + \sum_{t=0}^{k-1} A^t w_{k-t-1}. \tag{34}$$

In order to estimate $H_\tau$, post-multiply both sides of (34) with $\zeta_{k-\tau-1}^\top$ and rearrange the terms, and we get

$$x_k\zeta_{k-\tau-1}^\top = \left(\sum_{t=0}^{k-1} A^t w_{k-t-1} + A^k x_0\right)\zeta_{k-\tau-1}^\top + \sum_{t=0}^{k-1}(k-t)^{-\beta}H_t\zeta_{k-t-1}\zeta_{k-\tau-1}^\top +$$

$$\sum_{t=0}^{k-1} H_t\tilde{u}_{k-t-1}\zeta_{k-\tau-1}^\top. \tag{35}$$

To take the expectation of (35), notice that for any time step $k$, the variables $x_0, \zeta_0, \zeta_1, \ldots, \zeta_k$, $w_0, w_1, \ldots, w_k, \tilde{u}_0, \tilde{u}_1, \ldots, \tilde{u}_{k+1}$ are measurable w.r.t. $\mathcal{F}_k$, which, together with the independence among $x_0, \zeta_0, \zeta_1, \ldots, \zeta_k, w_0, w_1, \ldots, w_k$, leads to the following relations:

$$\mathbb{E}\left[\zeta_{k_1}\zeta_{k_2}^\top \mid \mathcal{F}_{k_2-1}\right] = \begin{cases} I & \text{if } k_1 = k_2 \\ 0 & \text{otherwise,} \end{cases} \tag{36}$$

$$\mathbb{E}\left[\tilde{u}_{k_1}\zeta_{k_2}^\top \mid \mathcal{F}_{k_2-1}\right] = \begin{cases} 0 & \text{if } k_1 \leq k_2 \\ \text{other values} & \text{otherwise,} \end{cases} \tag{37}$$

$$\mathbb{E}\left[w_{k_1}\zeta_{k_2}^\top \mid \mathcal{F}_{k_2-1}\right] = 0, \tag{38}$$

$$\mathbb{E}\left[x_0\zeta_{k_2}^\top \mid \mathcal{F}_{k_2-1}\right] = 0, \tag{39}$$

for any two time steps $k_1 \geq 0$ and $k_2 \geq 1$.

Now let us prove the conclusion using induction on $\tau$.

First consider the case $\tau = 0$: we have

$$\hat{H}_{k,0} - H_0 = \frac{1}{k}\sum_{i=1}^{k}\left[(i+1)^{\beta}x_i\zeta_{i-1}^{\top} - H_0\right] = \frac{1}{k}\sum_{i=0}^{k-1}\left[(i+2)^{\beta}x_{i+1}\zeta_i^{\top} - H_0\right]. \tag{40}$$

Let $\Phi_k = (k+1)^{\beta}x_{k+1}\zeta_k^{\top} - H_0$. By substituting (35) and (36) to (39) into (40) it can be seen $\mathbb{E}[\Phi_k \mid \mathcal{F}_{k-1}] = 0$, and hence $S_k \triangleq \sum_{i=0}^{k}\Phi_i$ is a martingale w.r.t. $\{\mathcal{F}_k\}$. Furthermore, we can verify $\mathbb{E}\left\|\Phi_k\right\|^2 \sim \mathcal{C}(2\beta)$: by Cauchy-Schwarz inequality,

$$\mathbb{E}\left\|(k+1)^{\beta}x_{k+1}\zeta_k^{\top}\right\|^2 \leq (k+1)^{2\beta}\sqrt{\mathbb{E}\left\|x_{k+1}\right\|^4}\sqrt{\mathbb{E}\left\|\zeta_k\right\|^4}.$$

By the procedure described in Algorithm 2, the switching threshold is no larger than $\log k$ for every step before $k$, and therefore, according to Lemma 9,

$$\mathbb{E}\left\|x_k\right\|^4 < 8\left[\mathcal{Q}(\log k, \rho, P) + \mathscr{W}^2\kappa(P)^2\right] \sim \mathcal{O}\left((\log k)^8\right) \sim \mathcal{C}(0),$$

where $0 < \rho < 1$ and $P \succ 0$ are constants such that $A^{\top}PA \prec \rho P$, which exist due to the stability of $A$, $\mathcal{Q}$ is defined as in Lemma 9, and $\mathscr{W}$ is defined in (25). Also taking note of the fact $\mathbb{E}\left\|\zeta_k\right\|^4 = p(p+2) \sim \mathcal{C}(0)$, we have

$$\mathbb{E}\left\|(k+1)^{\beta}x_{k+1}\zeta_k^{\top}\right\|^2 \sim \mathcal{C}(2\beta),$$

and hence,

$$\mathbb{E}\left\|\Phi_k\right\|^2 = \mathbb{E}\left\|(k+1)^{\beta}x_{k+1}\zeta_k^{\top} - H_0\right\|^2 \leq \mathbb{E}\left(\left\|(k+1)^{\beta}x_{k+1}\zeta_k^{\top}\right\| + \left\|H_0\right\|\right)^2$$

$$\leq 2\left(\mathbb{E}\left\|(k+1)^{\beta}x_{k+1}\zeta_k^{\top}\right\|^2 + \left\|H_0\right\|^2\right) \sim \mathcal{C}(2\beta).$$

By applying Lemma 11 to the martingale $\{S_k\}$ defined above, we get

$$\hat{H}_{k,0} - H_0 \sim \mathcal{C}\left(\beta - \frac{1}{2}\right).$$

Now assume that $\tau \geq 1$ and that we already have

$$\hat{H}_{k,t} - H_t \sim \mathcal{C}\left(\beta - \frac{1}{2}\right) \tag{41}$$

for $t = 0, 1, \ldots, \tau - 1$. Then

$$\hat{H}_{k,\tau} - H_{k,\tau}$$

$$= \frac{1}{k-\tau}\sum_{i=\tau+1}^{k}\left\{(i-\tau)^{\beta}\left[x_i - \sum_{t=0}^{\tau-1}\hat{H}_{k,t}\tilde{u}_{i-t-1}\right]\zeta_{i-\tau-1}^{\top} - H_{k,\tau}\right\}$$

$$= \frac{1}{k-\tau}\sum_{i=0}^{k-\tau-1}\left\{(i+1)^{\beta}\left[x_{i+\tau+1} - \sum_{t=0}^{\tau-1}\hat{H}_{k,t}\tilde{u}_{i+\tau-t}\right]\zeta_i^{\top} - H_{k,\tau}\right\}$$

$$= \frac{1}{k-\tau}\sum_{i=0}^{k-\tau-1}\left\{(i+1)^{\beta}\left[x_{i+\tau+1} - \sum_{t=0}^{\tau-1}H_t\tilde{u}_{i+\tau-t}\right]\zeta_i^{\top} - H_{k,\tau}\right\} -$$

$$\sum_{t=0}^{\tau-1}\left(\hat{H}_{k,t} - H_t\right)\frac{1}{k-\tau}\sum_{i=0}^{k-\tau-1}(i+1)^{\beta}\tilde{u}_{i+\tau-t}\zeta_i^{\top}. \tag{42}$$

Completely similarly to the case $\tau = 0$, we can show

$$\frac{1}{k-\tau}\sum_{i=0}^{k-\tau-1}\left\{(i+1)^{\beta}\left[x_{i+\tau+1} - \sum_{t=0}^{\tau-1}H_t\tilde{u}_{i+\tau-t}\right]\zeta_i^{\top} - H_{k,\tau}\right\} \sim \mathcal{C}\left(\beta - \frac{1}{2}\right).$$

Meanwhile, for each of $t = 0, 1, \ldots, \tau - 1$, define $\Phi_k^t = (k+1)^\beta \tilde{u}_{k-t} \zeta_k^\top$. In view of (37) it can be shown $\mathbb{E}[\Phi_k^t \mid \mathcal{F}_{k-1}] = 0$, and hence $S_k^t \triangleq \sum_{i=0}^k \Phi_k^t$ is a martingale w.r.t. $\{\mathcal{F}_k\}$. Furthermore, by the procedure described in Algorithm 2, $\|\tilde{u}_{k-t}\|^2 \le (\log(k-t))^4 \le (\log k)^4$, we have

$$\mathbb{E}\left\|\Phi_k^t\right\|^2 \le (k+1)^{2\beta} (\log k)^4 \, \mathbb{E}\left\|\zeta_k\right\|^2 \sim \mathcal{C}(2\beta).$$

By applying Lemma 11 to the martingale $S_k^t$ defined above, we get

$$\frac{1}{k-\tau} \sum_{i=0}^{k-\tau-1} (i+1)^\beta \tilde{u}_{i+\tau-t} \zeta_i^\top \sim \mathcal{C}\left(\beta - \frac{1}{2}\right),$$

which together with (41) implies

$$\left(\hat{H}_{k,t} - H_t\right) \frac{1}{k-\tau} \sum_{i=0}^{k-\tau-1} (i+1)^\beta \tilde{u}_{i+\tau-t} \zeta_i^\top \sim \mathcal{C}\left(2\left(\beta - \frac{1}{2}\right)\right) \sim \mathcal{C}\left(\beta - \frac{1}{2}\right).$$

Now that we have shown the RHS of (42) is the sum of $\tau + 1$ matrices, each of order $\mathcal{C}\left(\beta - \frac{1}{2}\right)$, we get

$$\hat{H}_{k,\tau} - H_{k,\tau} \sim \mathcal{C}\left(\beta - \frac{1}{2}\right).$$

According to our definition of $\mathcal{C}(\alpha)$, this is to say it holds almost surely that

$$\lim_{k \to \infty} \frac{\hat{H}_{k,\tau} - H_\tau}{k^{-\gamma+\epsilon}} = 0,$$

where $\gamma = 1/2 - \beta > 0$, for any $\epsilon > 0$, which concludes our proof. □

## A.4 PROOF OF THEOREM 3

Let us denote by $J_W^\pi$ the cost of a policy $\pi$ when acting on a system in the form (1) with process noise covariance $W$. Let us consider the following variants of policies:

- $\pi^*$: optimal policy, i.e., $\pi^*(x) = K^* x$, with $K^*$ defined in (6).

- $\hat{\pi}_k$: certainty equivalent policy with no exploratory noise, i.e., $\hat{\pi}_k(x) = \hat{K}_k x$.

- $\tilde{\pi}_k$: safe switching policy with no exploratory noise, i.e., Algorithm 2 invoked as $\pi(x, \xi; k, \hat{K}_k, +\infty)$.

- $\pi_k$: safe switching policy with exploratory noise, i.e., the actually applied policy at step $k$.

Notice that $J^{\pi_k} = J_W^{\pi_k}$ and $J^* = J_W^{\pi^*}$. Our plan is decomposing $J^{\pi_k} - J^*$ as

$$J^{\pi_k} - J^* = \left(J_W^{\pi_k} - J_{W+(k+1)^{-2\beta}I}^{\tilde{\pi}_k}\right) + \left(J_{W+(k+1)^{-2\beta}I}^{\tilde{\pi}_k} - J_{W+(k+1)^{-2\beta}I}^{\hat{\pi}_k}\right) +$$
$$\left(J_{W+(k+1)^{-2\beta}I}^{\hat{\pi}_k} - J_W^{\hat{\pi}_k}\right) + \left(J_W^{\hat{\pi}_k} - J_W^{\pi^*}\right),$$

and bounding the RHS terms respectively. We next tackle these terms in reverse order:

1. $J_W^{\hat{\pi}_k} - J_W^{\pi^*}$: for an arbitrary stabilizing linear feedback policy $\pi(x) = Kx$, we know the cost is $J_W^\pi = \mathrm{tr}(WP)$, where $P$ solves the discrete Lyapunov equation (32). By Theorem 2, the Markov parameter estimates converge as $\mathcal{C}(-\gamma)$. Since with random input in Algorithm 3, the matrix $\begin{bmatrix} \mathcal{U}^h \\ \mathcal{X}_0^h \end{bmatrix}$ is full row rank, it follows that the pseudo-inverse operator is differentiable, which together with Lemma 2 guarantees that $\hat{A}_k - A$, $\hat{B}_k - B \sim \mathcal{C}(-\gamma)$. Due to the differentiability of the stabilizing solution of the discrete algebraic equation, there is also $\hat{K}_k - K^* \sim \mathcal{C}(-\gamma)$. In particular, $\{\hat{K}_k\}$ converges to $K^*$ almost surely, and since $K^*$ is stabilizing, we have for almost every realization of randomness, $\hat{K}_k$ is also stabilizing for sufficiently large $k$. Assuming w.l.o.g. that $\hat{K}_k$ is stabilizng for any

$k$, we have $J_W^{\hat{\pi}_k} - J_W^{\pi^*} = \mathrm{tr}(W(P_k - P^*))$, where $P^*$ is the Lyapunov equation solution corresponding to $K^*$, also the solution to the Riccati equation (5), and $P_k$ is the Lyapunov equation solution corresponding to $\hat{K}_k$. According to Lemma 13, we have

$$\|P_k - P^*\|_F \le \left\| \left(I - \tilde{A}_k^\top \otimes \tilde{A}_k^\top\right)^{-1} \right\|_2 \|R\|_F \left\| \hat{K}_k - K^* \right\|_F^2,$$

where $\tilde{A}_k = A + B\hat{K}_k$. Since every $\tilde{A}_k$ is stable, we have $\left\| \left(I - \tilde{A}_k^\top \otimes \tilde{A}_k^\top\right)^{-1} \right\|_2$ is a continuous function of $\tilde{A}_k$. Meanwhile, $\{\tilde{A}_k\}$ converges to $A + BK^*$, which implies $\left\| \left(I - \tilde{A}_k^\top \otimes \tilde{A}_k^\top\right)^{-1} \right\|_2$ is bounded. Hence, $\hat{K}_k - K^* \sim \mathcal{C}(-\gamma)$ implies $P_k - P^* \sim \mathcal{C}(-2\gamma)$. Finally, we have $J_W^{\hat{\pi}_k} - J_W^{\pi^*} = \mathrm{tr}(W(P_k - P^*)) \sim \mathcal{C}(-2\gamma)$.

2. $J_{W+(k+1)^{-2\beta}I}^{\hat{\pi}_k} - J_W^{\hat{\pi}_k}$: from $J_W^{\hat{\pi}_k} = \mathrm{tr}(WP_k)$, with $P_k$ defined the same as above, we have $J_{W+(k+1)^{-2\beta}I}^{\hat{\pi}_k} - J_W^{\hat{\pi}_k} = \mathrm{tr}((k+1)^{-2\beta}P_k)$. Since $\{P_k\}$ converges to $P^* \succ 0$, we have $J_{W+(k+1)^{-2\beta}I}^{\hat{\pi}_k} - J_W^{\hat{\pi}_k} \sim \mathcal{C}(-2\beta)$.

3. $J_{W+(k+1)^{-2\beta}I}^{\tilde{\pi}_k} - J_{W+(k+1)^{-2\beta}I}^{\hat{\pi}_k}$: we can basically apply Lemma 10. To verify the conditions of Lemma 10, we first fix $\rho, P$: since $A + BK^*$ is stable, for any fixed $Q \succ 0$, the discrete Lyapunov equation $(A + BK^*)^\top P(A + BK^*) + Q = P$ has a solution $P \succ 0$. Therefore, there exists $P \succ 0, 0 < \rho < 1$, such that $(A + BK^*)^\top P(A + BK^*) < \rho P$. Since $\{\hat{K}_k\}$ converges to $K^*$ almost surely, we may assume w.l.o.g. that $(A + B\hat{K}_k)^\top P(A + B\hat{K}_k) < \rho P$ for any $k$. Furthermore, since $A$ is stable, there is $A^t \to 0$ as $t \to \infty$, and therefore, with $t = \lfloor \log k \rfloor$ sufficiently large, we also have $(A^t)^\top P A^t < \rho P$. For an upper bound of the noise magnitude, we insert $W + I$ in place of $W$ in (25) to compute $\mathcal{W}$. Now that $A, B, Q, R, \mathcal{W}, \rho, P$ are all fixed, we can apply the conclusion of Lemma 10 by taking the limit $\hat{K}_k \to K^*$ to obtain $J_{W+(k+1)^{-2\beta}I}^{\tilde{\pi}_k} - J_{W+(k+1)^{-2\beta}I}^{\hat{\pi}_k} \sim \mathcal{O}((\log k)^2 \exp(-c(\log k)^2)) \sim \mathcal{C}(-\infty)$.

4. $J_W^{\pi_k} - J_{W+(k+1)^{-2\beta}I}^{\tilde{\pi}_k}$: these two costs are associated with the same closed-loop system and differ only in the $u_k^\top R u_k$ terms. In particular, $J_W^{\pi_k} - J_{W+(k+1)^{-2\beta}I}^{\tilde{\pi}_k} = \mathrm{tr}((k+1)^{-2\beta}R) \sim \mathcal{C}(-2\beta)$.

Adding up the above four terms using Lemma 2, we obtain $J^{\pi_k} - J^* \sim \mathcal{C}(\max\{-2\beta, -2\gamma, -\infty\}) \sim \mathcal{C}(-\min\{2\beta, 2\gamma\})$. According to our definition of $\mathcal{C}(\alpha)$, this is to say it holds almost surely that

$$\lim_{k \to \infty} \frac{J^{\pi_k} - J^*}{k^{-\min\{2\beta, 2\gamma\}+\epsilon}} = 0,$$

for any $\epsilon > 0$, which concludes our proof.

## B  AN ILLUSTRATION OF OSCILLATION UNDER SWITCHING

In this section, we provide a simple illustrative example to explain why we need prolonging "non-action" period in Algorithm 2.

Consider a simple two-dimensional noise-free system

$$x_{k+1} = A_k x_k,$$

where the candidates for $A_k$ are

$$A_0 = \begin{bmatrix} 0.5 & 2 \\ 0 & 0.5 \end{bmatrix}, A_1 = \begin{bmatrix} 0.5 & 0 \\ 2 & 0.5 \end{bmatrix}.$$

It can be seen $\rho(A_0) = \rho(A_1) = 0.5$, i.e., both the system matrices are stable.

Now consider the following switching strategy:

- If $\|x_k\| \geq M$, then apply $A_0$ for $t$ consecutive steps.
- Otherwise, apply $A_1$.

Figure 4 shows simulation results with $x_0 = (0.1, 1)$, $M = 1$ and $t = 1, 2$. We can observe that even for this simple system, the frequent switching caused by $t = 1$ may cause the state to oscillate, while $t = 2$ suffices to suppress the oscillation. Indeed, we can verify $\rho(A_1 A_0) \approx 4.5 > 1$ even though both $A_0$ and $A_1$ are stable. However, as long as $A_0$ is stable, $A_1 A_0^t$ will eventually become stable as $t$ is chosen to be sufficiently large. This explains why we use prolonging $t$ in our policy.

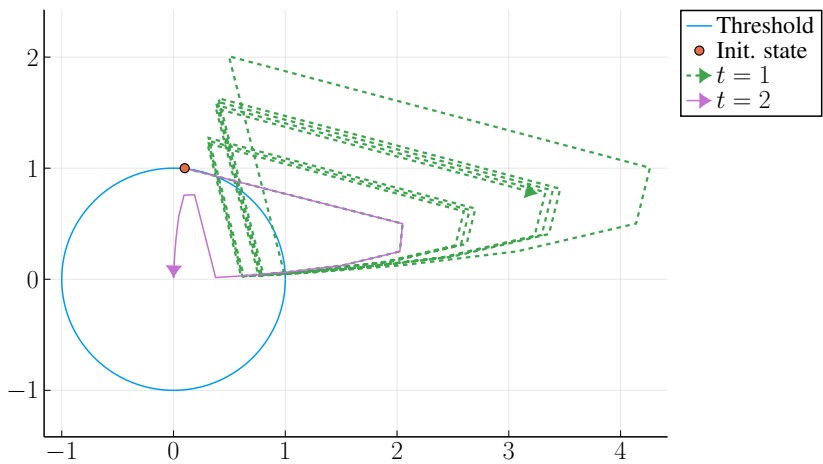

Figure 4: Illustrative example of oscillation under switching. In this example, $t = 1$ causes the state to oscillate, while $t = 2$ can suppress the oscillation.

