# OpenReview forum: "Safe Linear-Quadratic Dual Control with Almost Sure Performance Guarantee"
_ICLR.cc/2022/Conference — ICLR 2022 Submitted_

### Official Review · Reviewer_1scp · 2021-11-02

**Correctness:** 3
**Technical Novelty And Significance:** 3
**Empirical Novelty And Significance:** Not applicable
**Recommendation:** 6
**Confidence:** 5

**Main Review:**

Update: After the edits the authors did, together with the edits they promise to do, the paper can be accepted.
_________________________________________________________
Original Review: Regretfully, the manuscript cannot be accepted due to different issues. First, it aims to re-establish existing results. Moreover, there are frequent incorrect and/or incomplete claims in the manuscript. Unfortunately, lack of novelty, lack of connections to existing work, incomplete literature review, unsupported and/or wrong statements, and subpar presentation, prevent the manuscript from being accepted.

The claim that high probability results do not imply almost sure convergence is ONLY true if the failure probability is a design parameter. (Of course, even in this case, by letting it shrink as time grows, we obtain almost sure convergence). So, to see a work that failure probability is not a used in the algorithm, search for "Input perturbations for adaptive control and learning" and see the reference therein. Specially, one of the references exactly establishes the almost sure results this manuscript aims to provide. Further, in the existing literature, the problem is addressed without the assumption of open-loop stability (or initial stabilizer), and I recommend the authors to use google scholar to find the rich literature of adaptive stabilization.

Here, I mention a few more issues, but there are multiple similar things the authors need to address for resubmitting the manuscript. The existing literature also relaxes the assumptions on the noise so that neither normality nor stationarity are needed. Also, there are results on worst-case performance (ie stochastic sub-optimality) which are stronger than the approach of average-case (ie expected) sub-optimality of the manuscript. I guess by "learning process" in Def 2 the authors mean 'sequence of policies'. It is unclear why line 7 of algorithm 1 is better than the least-squares estimate (although it is very similar). About Lem 10, isn't it an immediate consequence of sub-Gaussianity? Statement of Lemma 1 assumes full accuracy, and so is unacceptable and does not provide consistency (as the authors claim). "It should be noticed ..." on p8 is not accurate. Finally, "TS is in general computationally inefficient" needs justifications.



**Summary Of The Paper:**

The paper studies reinforcement learning policies for an unknown linear system with quadratic cost functions. It is shown that the presented algorithm provides consistent estimates and the rates are provided.

**Summary Of The Review:**

Update: After the edits the authors did, together with the edits they promise to do, the paper can be accepted.
_________________________________________________________
Original Summary: Regretfully, the manuscript cannot be accepted due to different issues. First, it aims to re-establish existing results. Moreover, there are frequent incorrect and/or incomplete claims in the manuscript. Unfortunately, lack of novelty, lack of connections to existing work, incomplete literature review, unsupported and/or wrong statements, and subpar presentation, prevent the manuscript from being accepted.

---

> ### Author Response · Authors · 2021-11-20
> **Response to Reviewer 1scp**
>
> Thank you for the serious criticisms and suggestions. We admit having omitted a few relevant references, which we have added in the revised version, but we respectfully disagree with your opinion that we are re-establishing existing results. We address the specific comments below:
>
> - > By letting the failure probability shrink as time grows, we obtain almost sure convergence
>
>   Yes, it is well known that in principle we can make the failure probability shrink and establish almost sure convergence using Borel-Cantelli lemma, but in the online setting, how can we "let the probability shrink as time grows", considering the system state may have already exploded? We did not find existing work that implements this method in an online setting. Could you enlighten us by pointing to some specific references?
>
> - > One of the references in "Input perturbations for adaptive control and learning" exactly establishes the almost sure results this manuscript aims to provide
>
>   We are not sure whether you mean "On Adaptive Linear-Quadratic Regulators" (Faradonbeh et al. 2020b). Thank you for pointing us to this insightful reference, which we have cited in the revised paper, but the almost-sure results therein are conditioned upon the claim "we assume that ... and the systems remains stable when RCE or TS is being employed" at the end of Sec 4.2, an assumption that neither holds naturally nor can be guaranteed a.s. by adaptive stabilization. By contrast, our proposed safeguard scheme exactly addresses the potential destabilization of the system, and guarantees that the estimation error and suboptimality gap converge even if destabilizing feedback gains are learned from data in the process.
>
> - > The problem is addressed without the assumption of open-loop stability (or initial stabilizer), and I recommend the authors to use google scholar to find the rich literature of adaptive stabilization
>
>   We believe adaptive stabilization and online LQR are related but different problems. We have pointed to adaptive stabilization as a compelling method for finding the initial stabilizer, at the end of Section 2 of the revised paper. Afterwards we assume the existence of an initial stabilizer, as was done previous works on online LQR, (Mania et al., 2019; Simchowitz & Foster, 2020), etc.
>
> - > The existing literature also relaxes the assumptions on the noise so that neither normality nor stationarity are needed
>
>   Yes, we are aware of this point, and straightforward extensions can be made to our method for different noise models. However, we choose to exclude these extensions from the paper, because they are less relevant to our main contribution (the safeguard scheme that guarantees bounded-cost safety and a.s. convergence even with potentially destabilizing gains).
>
> - > There are results on worst-case performance (ie stochastic sub-optimality) which are stronger than the approach of average-case (ie expected) sub-optimality of the manuscript
>
>   There are different performance metrics, e.g., in Faradonbeh et al. (2020a). The average-case cost we consider, however, is the default objective of LQR, which is completely standard in both the classical formulation and in recent literature on online LQR, e.g., Mania et al., (2019); Simchowitz & Foster, (2020).
>
> - > I guess by "learning process" in Def 2 the authors mean 'sequence of policies'
>
>   Yes. This has been clearly stated just above Def 2.
>
> - > It is unclear why line 7 of algorithm 1 is better than the least-squares estimate (although it is very similar)
>
>   We never claim our method is strictly better, and it is just an alternative method to standard least-squares. We have added a discussion of this issue in Remark 3 of the revised paper.
>
> - > About Lem 10, isn't it an immediate consequence of sub-Gaussianity?
>
>   We do not see how Lemma 10 can be trivial. A large part of Appendix A.1 is devoted to establishing a rigorous proof of this result.
>
> - > Statement of Lemma 1 assumes full accuracy, and so is unacceptable and does not provide consistency (as the authors claim)
>
>   We have stated here by consistency we mean "accurate Markov parameter estimates would generate an optimal exploitation input". We did not base our main results on any unproofed claim.
>
> - > Inaccurate claims in related works
>
>   Thank you for pointing out. We have revised the statements accordingly.
>
>
> References:
>
> Mohamad Kazem Shirani Faradonbeh, Ambuj Tewari, and George Michailidis. Input perturbations for adaptive control and learning. Automatica, 117:108950, 2020a.
>
> Mohamad Kazem Shirani Faradonbeh, Ambuj Tewari, and George Michailidis. On adaptive linear–quadratic regulators. Automatica, 117:108982, 2020b.
>
> Horia Mania, Stephen Tu, and Benjamin Recht. Certainty equivalence is efficient for linear quadratic control. arXiv preprint arXiv:1902.07826, 2019.
>
> Max Simchowitz and Dylan Foster. Naive exploration is optimal for online lqr. In International Conference on Machine Learning, pp. 8937–8948. PMLR, 2020.

---

> ### Author Response · Authors · 2021-11-24
> **Further explanations on stability vs. optimality, and our novelties**
>
> We thank the reviewer for the very timely reply. We would like to give some further explanations on the issues you mentioned, to make sure we are on the same page:
>
> 1. Brief summary of how our scheme works:
>
>     Our policy is the certainty equivalent (CE) policy with an additional safeguard, which modifies the policy to "take no action" for $t_k\sim O(\log k)$ consecutive steps whenever the norm of either the current state or the CE feedback gain exceeds a threshold $M_k\sim O(\log k)$, where $k$ is the step index. Random perturbations are also added to the above modified CE control input for exploring the system parameters. Roughly speaking, whether the CE feedback gains are stabilizing or not during the learning process, the safeguard mechanism clamps the state at $\tilde{O}(1)$ (Lemmas 7,8,9), which ensures that the parameter estimates converge to the true parameters almost surely at $\tilde{O}(k^{-1/4})$ (Theorem 2). Therefore, for almost every trajectory, the feedback gains will eventually become stabilizing and converge to the optimal one. Furthermore, since the norm of the state is extremely unlikely to exceed $M_k$ as long as the learned feedback gain becomes stabilizing and as $M_k$ increases over time, we can prove the suboptimality caused by the conservativeness of the safeguard mechanism is only $\tilde{O}(1)$ (Lemma 10). By combining the three sources of suboptimality, i.e., estimation error, random perturbations and conservativeness of safeguard, we can finally proof that the suboptimality gap of policies converge to zero almost surely at $\tilde{O}(k^{-1/2})$ (Theorem 3).
>
> 2. Why instability matters:
>
>     We mention instabilities in our explanations, because there is always a nonzero probability that a destabilizing CE *feedback gain* is learned from *finite* data. The role of our safeguard mechanism is to make sure that the *policy* has *bounded cost* (not necessarily making the closed-loop system ISS stable), even if the feedback gain is destabilizing. This is achieved by modifying CE to a nonlinear switched policy (Algorithm 2) parameterized by the learned feedback gain. The switching mechanism is essential for the a.s. convergence of our algorithm, which means destabilizing feedback gains will no longer occur as the time *tends to infinity*. The aforementioned a.s. convergence property differs with existing works on online LQR because:
>
>     - In works that provide with-high-probability performance guarantees, it is typical that the learned feedback gain at some time step is stabilizing with probability at least $1-\delta$, but there is no performance guarantee for the complementary event that holds with probability up to $\delta$.
>     - The work "On Adaptive Linear-Quadratic Regulators" (Faradonbeh et al. 2020b), which provides a.s. guarantees under the assumption of stable closed-loop systems, does not consider the situation where destabilizing feedback gains are learned from data, a situation that may occur even if there exists a known set of stabilizers.
>
>    By contrast, the switching mechanism we propose allows us to establish a.s. convergence properties without assumptions beyond open-loop stability or existence of one initial stabilizer.
>
> 3. On stabilization vs. optimality:
>
>     Frankly we are a little confused as to where we "denied relevance of stabilization to optimality". For a policy, stabilization is necessary but not sufficient for optimality. As far as we are concerned, adaptive stabilization is focused on stabilizing an unstable system, while online LQR tries to find the optimal policy, typically after pre-stabilization, and in this sense they can be viewed as complementary aspects of the linear adaptive control problem. Please kindly point out if anything is missing.
>
>     With regard to our proposed scheme, it is possible that destabilizing feedback gains are learned from data, even after pre-stabilization, and this is why we need the safeguard/switching mechanism detailed above. Clearly the policy at a *fixed* time step is suboptimal due to the conservativeness of the safeguard, which can be seen as the cost of preventing potential destabilization. However, as time tends to infinity, the overall suboptimality gap will *converge* to zero, at a preferable rate, as is stated and proved in Theorem 3.
>
> 4. Our contributions and novelties
>
>     As a TLDR: we propose a safeguard added to CE, which guarantees a.s. convergence of both estimation error and policy suboptimality gap, without assumptions beyond open-loop stability or existence of a known initial stabilizer. Such guarantee was not present in previous works. Also, the closed-loop system under our scheme is nonlinear due to the switching mechanism, whose analysis differs from existing works on learning LQR.
>
> Please let us know if you have further questions.

---

### Official Review · Reviewer_oE9i · 2021-11-03

**Correctness:** 4
**Technical Novelty And Significance:** 3
**Empirical Novelty And Significance:** 3
**Recommendation:** 8
**Confidence:** 4

**Main Review:**

Although my overall impression of the paper is positive, there are several concerns which limit the significance of the result. First, while the definition of a "safe learning process" is clearly articulated in Definition 2, such a definition by itself does not ensure stability, as the act of switching between stable policies may be unstable -- as observed by the authors. While the authors address this issue using inherent stability of the process and implementing what is effectively a kill switch, such an approach is clearly conservative, and, moreover, there is no statement in the main results directly establishing stability of the trajectories (although clearly such trajectories are bounded).

This brings up the second point, which is that the proposed approach is rather cautious. Convergence in the simulation seems to require $~10^8$ time steps - presumably due to the rather large state-dimension of the numerical example.

Another point which I found significant is the omission of the vast literature on the problem of adaptive control (aside from the 1960 paper). While clearly the adaptive control approach is dated and only estimates a few system parameters, it considers precisely the problem formulated by the authors.

Finally, I would have liked to see more justification for the estimate of the performance metric in the numerical analysis. It is unclear to me that Algorithm 2 actually produces a policy -- as opposed to simply the next input value. Yet computing the objective requires such a policy. What is being done here?

If the issues mentioned above can be satisfactorily addressed, the paper should be suitable for publication.

Notes:
- Why is there no $B_2$ matrix for representing the effect of process noise?
- It is unclear if the numerical examples include process noise and if so, what are the properties of this noise?
- The simulated is system is non-trivial and provides plausible insight into the performance of the algorithm.
- Remark 2: Can the authors support "easily satisfied" with a proof or reference?
- The use of $J^{\pi_k}$ is a bit unclear. Algorithm 2 does not actually return a policy -- only the next input and internal state. How can a policy be extracted from this algorithm? -- a policy which is needed to compute the performance index.
- Define "almost surely" in Theorem 2.
- Typos: "The complete algorithm we propose LQR dual control is presented in Algorithm 1."; "would suffice characterize"; ""


**Summary Of The Paper:**

This paper addresses the problem of combining system identification and optimal control in an online framework for stable linear systems with full-state sensing. Despite the restrictions of linearity, stability, and full-state sensing, the proposed problem formulation is still challenging and I am unaware of any results which provide the same optimality guarantees provided by the authors in the online framework. The paper itself is unusually well-conceived, well-written, well-structured, is clear, and makes very efficient use of notation.

**Summary Of The Review:**

My overall impression of the paper is positive. While the switching between policies seems a bit ad-hoc, the performance and convergence guarantees are solid -- correctly formulated and true. The paper addresses a real problem of current interest and provides a solid theoretical foundation for the proposed method. The numerical simulations are non-trivial and convincing. I have some minor concerns, but they can likely be addressed in the author response.

---

> ### Author Response · Authors · 2021-11-20
> **Response to Reviewer oE9i**
>
> Thank you very much for your encouraging comments and constructive suggestions. We address the specific comments below:
>
> - Definition 2 does not ensure stability
>
>   We totally agree that our definition of safety does not imply stability. We have changed the term to "bounded-cost safety" in the revised paper, which would hopefully reveal the essential property of our safeguard scheme. The stability of trajectories of the nonlinear and time-varying closed-loop system under our scheme is an interesting and challenging topic under investigation.
>
> - The proposed approach is rather cautious
>
>   Theoretically, we have pointed out in the main context that the additional cost incurred by the cautiousness of our approach contributes only $\tilde{o}(1)$ to the overall policy suboptimality gap, whose complete proof is given in the appendix (Lemma 10 and proof of Theorem 3). Empirically, we have added an ablation experiment in the revised paper, which demonstrates that our proposed approach, while preventing catastrophic bad cases, does not harm the performance in good cases, compared to certainty equivalence.
>
> - Omission of the literature on adaptive control
>
>   Thank you very much for the valuable suggestion. We have added references to classical adaptive control in the revised version.
>
> - Justification for the estimate of the performance metric
>
>   Algorithm 2 itself, invoked with certain parameters, is a policy. We have further clarified this point in the revised version.
>
> - Why is there no $B_2$ matrix for representing the effect of process noise?
>
>   We assume the process has a non-trivial covariance matrix $W$ in our formulation, which can absorb linear transformations on the noise. Therefore we can assume w.l.o.g. $B_2$ is identity.
>
> - Other minor comments
>
>   Thank you very much for the detailed and specific suggestions. We have correspondingly updated the paper.

---

> > ### Comment · Reviewer_oE9i · 2021-11-24
> > **Policies?**
> >
> > Generally, I think the authors did a fair job of addressing the reviewer responses, although with some prodding in one case. However, I would still like clarification on the one point:
> >
> > Algorithm 2 does not actually return a policy -- only the next input and internal state. How can a policy be extracted from this algorithm? -- a policy which is needed to compute the performance index

---

> > > ### Author Response · Authors · 2021-11-24
> > > **Clarification on policies**
> > >
> > > Thank you very much for the response!
> > >
> > > Please notice that Algorithm 2, when instantiated with fixed parameters $k$, $K$ and $\beta$, is a function that takes in state $x$ and policy internal state $\xi$, and outputs control input $u$ and next internal state $\xi'$, i.e., a policy. In line 10 of Algorithm 1, we define $\pi_k$ as the subroutine described by Algorithm 2, instantiated with parameters $k, \hat{K}_k, \beta$, and this $\pi_k$ is the policy used for computing the final performance index.
> > >
> > > We believe this is a presentation, rather than logical issue. Sorry for the confusion, and hopefully some minor rewordings in the final version will make this point clearer.
> > >
> > > Thank you again for your valuable advise!

---

### Official Review · Reviewer_qNrU · 2021-11-04

**Correctness:** 3
**Technical Novelty And Significance:** 3
**Empirical Novelty And Significance:** 3
**Recommendation:** 5
**Confidence:** 4

**Main Review:**

The question asked by the paper is very intriguing: establishing a.s. guarantees instead of w.h.p. guarantees for learning-based control is indeed very important for safe implementation on physical systems. Hence, I read the paper with great interest and checked the proofs of the paper. The paper is well-written. However, I have several concerns listed below.

1. [Def 2] My biggest concern on the safety definition is on Def 2. In this paper, a time-varying sequence of policies is considered, thus inducing a time-varying system. It is well-known that the stability of individual time-invariant systems does not guarantee the stability of the time-varying systems, even for linear systems, i.e. even if Ak is stable for all k, the linear time-varying system x_{k+1}=Ak x_k can no longer be stable. In fact, this is one of the major difficulties in adaptive control, i.e. how to guarantee stability while updating the policies. One approach to get around this is by considering piecewise-constant policies (or episodic based policies) as in most papers on learning-based LQR. But this paper updates the policy at every step, which calls for other ways to ensure closed-loop stability (e.g. slowly varying policies), hence I found it weird to define stability/safety purely on individual policies.

2. [Def 1 and Theorem 1] Further, for Def 1, the definition makes sense for linear systems because a finite stochastic LQR's cost happens to coincide with the stability of a linear policy u=Kx. However, the policy designed in Algo 2 is essentially a nonlinear policy or piecewise linear policy. For nonlinear policies, a more common stability notion is input-to-state stability (ISS), which essentially requires that if the noises are small, then the states will be small. However, Def 1 does not seem to capture this. Further, Def 1 does not seem to require the stability of K used when |x|< log(k) for some index k in the policy structure considered in Algo 2. This suggests that a safe policy according to Def 1 may constantly steer the state to a distant though bounded location even when the system noises w are very small. Hence, I found the safe definition in Def 1 quite weak and it can be satisfied by merely adding this switch-to-0 policy structure without implying any stability property on the system.

3. [Algo 3 and Theorem 2] The Markov parameter estimation is very interesting. But I hope the authors can provide more explanation on why using Markov parameter estimation instead of the traditional least-square estimation directly on A, B. For output-feedback, it is more intuitive to learn markov parameters, but for full-state-feedback, learning markov parameters seem to increase the number of parameters to be learned, thus increasing the sample complexity and slower the learning. The only reason I can imagine from reading the paper is that the authors are able to establish the a.s. convergence rate of Alg 3, while the current literature on least-square estimation usually gives results on finite-sample confidence bound with some probability p<1, e.g. [Dean et al. 2018]. That being said, those literature focuses on the finite-sample case, but this paper considers the asymptotic case, so I am struggling to compare the results in this paper and in the literature. From one side, since ordinary least square also has strong consistency, so least square estimation should also result in a.s. convergence in the asymptotic sense, right? From the other side, how does Theorem 2 extend to the non-asymptotic case? I guess the result will still look like confidence bound with some probability in (0,1)? I would really love to hear the authors' thoughts on this.

4. [Theorem 3] The authors compare Theorem 3 with the regret analysis in (Simchowitz & Foster (2020)), but I found this comparison not straightforward. Notice that J^pi_k does not represent the stage cost, but the infinite-horizon-averaged cost by implementing the policy generated at time k in a separate trajectory for infinitely long. So summing J^pi_k over k does not directly represent the realized cost of the adaptive learning algorithm. Could the authors provide more comments on this difference from the standard regret guarantee?

5. [Simulation] I appreciate the numerical results provided in this paper. I just wonder how this algorithm compares with existing adaptive learning algorithms? it would be great to show that some scenarios where some existing adaptive learning algorithm does not converge but your algorithm does.


**Summary Of The Paper:**

This paper studies an adaptive control for LQR and provides an algorithm to converge to the optimal policy almost surely in an asymptotic sense, where the convergence rate is also provided. The paper assumes that the system is open-loop stable, and switches to zero control input (with exponentially decaying excitation noises) for a while when the state or control gain is larger than O(log(k)), which can steer the state to a smaller neighbourhood of 0 due to the open-loop stability of the original system. Theoretically, the paper shows that any switched policy generated by the algorithm guarantees a finite infinite-horizon-averaged cost, which is called "safe" in this paper. Further, the paper establishes the strong consistency of the Markov parameter estimation in Alg 3 with an error convergence rate. Besides, the paper shows that the learned policies converge to the optimal policy at a rate of O(1/sqrt(k)) in terms of the infinite-horizon-averaged cost. Lastly, the performance is evaluated with different parameter choices of the algorithm.

**Summary Of The Review:**

The paper asks a very important question: establishing a.s. guarantees instead of w.h.p. ones for safety. The paper is well-written. However, the safety definitions introduced in this paper do not indicate the stability of the system in any sense, thus I found these safety requirements relatively weak. Further, though the authors compare the results with the regret bounds, Theorem 3's result does not directly reflect the regret, so I hope to see more comments on the connections between Theorem 3 and regret. Lastly, the paper establishes a.s. guarantees only in the asymptotic sense, instead of for finite samples. Since the least square estimation used in the current literature also converges a.s. asymptotically, I am not sure how much progress this paper provides compared with the literature. I think it would be much more interesting to establish a.s. guarantees in non-asymptotic cases (some random example that comes to me right now is  Regret <o(T) a.s. for any finite T, instead of w.h.p., but any other a.s. guarantees would be great, especially in constraint satisfaction, which is also an important requirement for safety).

---

> ### Author Response · Authors · 2021-11-20
> **Response to Reviewer qNrU**
>
> Thank you very much for your detailed comments and constructive suggestions. Here we address the comments:
>
> - Definition of safety
>
>   We agree that our definition of safety does not imply the stability of the system. What our safeguard scheme does is actually preventing the state from exploding exponentially, and hence ensuring the convergence of estimation error and performance suboptimality gap. We totally agree with you that to prove the stability of trajectories (and regret analysis) would probably require slowly varying or episode-based policies, but results on stability or regret are still under investigation. We would like to point out respectfully that the nonlinear closed-loop system and non-Gaussian states under our scheme requires different analysis techniques from existing works on learning-based LQR, and we believe the currently presented results are non-trivial.
>
> - Def 1 and Theorem 1
>
>   Yes, our scheme does not ensure ISS stability, and the state can be driven to a bounded location given K is destabilizing. This is exactly the expected behavior, because it is always possible that a destabilizing K is learned from noisy data, and the purpose of our safeguard scheme is clamping the state at a bounded location, such that the system does not break down and the parameter estimator can converge almost surely. As sufficient data is collected, however, we have proved that K will eventually converge to the optimal gain and the state will no longer be driven away from zero.
>
> - Algo 3 and Theorem 2
>
>   It is possible to replace our parameter estimator with least-squares, which will give exactly the same almost-sure convergence rate, as we have clarified in Remark 3 of the revised paper. Here we propose an alternative scheme whose extensions to partially observed LQG setting and heavy-tailed noise models are straightforward. We believe a non-asymptotic analysis of our method is also an interesting topic for further investigation.
>
> - Theorem 3
>
>   Thank you for pointing out potential confusions. Theorem 3 (convergence rate of suboptimality gap) does not directly give any guarantee for regret, and what we compare against is the (high-probability) bound of suboptimality gap in Simchowitz & Foster (2020). We have further clarified this point in Remark 4 of the revised paper. As we mentioned in the first point, regret analysis of the nonlinear system under our scheme is still an interesting and challenging topic under investigation.
>
> - Simulation
>
>   Thank you very much for the suggestion. We have compared our scheme with naive certainty equivalence in the simulation section. In particular, certainty equivalence has a significant chance of divergence, and we argue that the failure probability is always nonzero with existing algorithms.
>
> - > I think it would be much more interesting to establish a.s. guarantees in non-asymptotic cases (e.g., Regret <o(T) a.s. for any finite T)
>
>    Surely those kind of results would be very interesting, but we argue that non-asymptotic a.s. guarantees would be extremely challenging, if not impossible, at least with the common notion of estimation error, policy suboptimality gap, or regret. This is because given a finite-size dataset corrupted by random noise, there is always a nonzero probability of getting "very unlucky". It can be expected that the non-asymptotic properties of our algorithm would hold with probability $1-\delta$ just as existing results, but what distinguishes our algorithm from existing ones is the asymptotic convergence guarantee. We will work on looking for new definitions that will better characterize the merit of our method in terms of finite-time performance.

---

### Official Review · Reviewer_ZdU6 · 2021-11-06

**Correctness:** 3
**Technical Novelty And Significance:** 3
**Empirical Novelty And Significance:** 3
**Recommendation:** 5
**Confidence:** 3

**Main Review:**

The studied problem of online learning for LQR control is a very interesting topic and has gained significant attention lately. There are many existing results on the regret analysis in a probabilistic setting, e.g. using certainty equivalent control strategy. Compared to that line of work, the key difference in this paper is the modified version of the standard certainty equivalent control strategy, by allowing the system to take zero action for a certain amount of time and avoid potential destabilization. While the reviewer found this idea intriguing, there are places where further clarification is needed to better justify the claims. Please see detailed comments as follows:

- Safety v.s. stability: it is a bit confusing to use the notion of safety rather than stability in the paper, since safe learning often refers to a different problem where a dynamical system is expected to avoid entering certain risky states during learning. The reviewer would suggest to use stable/stability that better describe the discussed problem here.

- Motivation of using Markov parameter inference: while the convergence of parameter inferences looks good, it is unclear why not use the standard least square estimation that directly estimate the matrices A and B. Or in other words, will it disrupt performance and estimation guarantees if switching to the least square approach?

- Proof of theorem 1: One key claim is that all policies derived from the proposed learning process is stable, which seems to mainly rely on the assumption of the innate stability of the system without external control input. However, it is difficult to see how this could be justified given the switching control strategy which also has the normal certainty equivalent control component with exploration noise. For example, it seems the proof of Theorem 1 (Section A.2) only discusses the innate stability analysis when \norm{x_i} is larger than log k. Also, given the assumption of the innate stability with the matrix A, wouldn’t that be straightforward to simply take no-action and thus converging to the origin easily?

- Correspondence to optimal rates with high probability: While it is claimed that the derived performance match up with the known optimal regret in the high probability regime, the reviewer does not find the analysis. Authors are encouraged to provide more details in the main context to help build the connections. For example, it is still unclear to me how the almost sure performance is achieved despite the stochastic noise in the dynamics.

- Experimental results: there are no comparison results to other work (e.g. standard certainty equivalent control), which is a weakness to the reviewer.

===== After Rebuttal ======
I appreciate the authors’ responses and revised content. Given the remaining concerns from the reviewer, additional feedback has been provided in the detailed response to help strengthen and justify the technical contribution further.


**Summary Of The Paper:**

This paper addresses the unconstrained stochastic linear-quadratic dual control problem with online parameter identification in an online setting, where a near-optimal control policy is sought to minimize the infinite-horizon quadratic cost (i.e. stabilizing the linear dynamical system) while learning the initially unknown matrices A and B. Instead of deriving a regret bound with a high probability done in the standard certainty equivalent learning schemes, this paper focuses on the almost sure convergence to the optimal control performance. The presented main contribution is the proposed switched control strategy (a stability-augmented certainty equivalent controller) and a parameter inference scheme that estimates the impulse response of the dynamical system in contrast of the matrices A and B. Theoretical analysis of the convergence rate of inference error and control performance is provided. Simulation results on the Tennessee Eastman Process (TEP) are given to validate the proposed LQ dual control algorithm.

**Summary Of The Review:**

The paper is in general well-written, but the reviewer still has several critical concerns over the justification of the claimed property due to lack of enough details. Authors are encouraged to provide more discussion in the main context to justify how the proposed components lead to the improvement compared to the standard approaches. More comparative results to other baseline algorithms are necessary to demonstrate the claimed significance of the proposed algorithm.

---

> ### Author Response · Authors · 2021-11-20
> **Response to reviewer ZdU6**
>
> Thank you very much for your detailed comments and constructive suggestions. Here we address the comments:
>
> - Safety v.s. stability
>
>   Thank you for pointing out the confusion about the notion of safety. We have changed the term to "bounded-cost safety" in the revised paper, which would hopefully reveal the essential property of our safeguard scheme. To further clarify, our notion of "bounded-cost safety" is different from the common notion of stability, i.e., input-to-state (ISS) stability, which requires that the state is small whenever the noise is small. In fact, during the process of any learning scheme, it is always possible that a destabilizing linear feedback gain is learned from data, in which case it is too stringent to require ISS stability at each step. What our safeguard mechanism does is preventing the state from exploding (as suggested by "bounded-cost"), and making sure the algorithm converges when the time goes to infinity.
>
> - Motivation of using Markov parameter inference
>
>   Least-squares will also converge, and provide the same asymptotic estimation and performance guarantees. The Markov parameter inference can be seen as an alternative method whose extensions to partially observed LQG setting and heavy-tailed noise models are straightforward, as is suggest by Remark 3 in the revised paper.
>
> - Proof of theorem 1
>
>   Yes, "bounded-cost safety" only requires the transition matrix of the system is (uniformly) stable when the state norm exceeds a fixed threshold, as is suggested by Lemma 7. Of course the "taking no action" policy is also "bounded-cost safe", but the suboptimality gap would not converge.
>
> - Correspondence to optimal rates with high probability
>
>   By "match-up" we mean: under the regret-optimal algorithm proposed by Simchowitz & Foster (2020),  both parameter estimation errors and policy suboptimality gaps scale at $\tilde{O}(1/\sqrt{T})$ with high probability, and under our algorithm, both the components scale at $\tilde{O}(1/\sqrt{T})$ almost surely. This point has been further claried in Remark 4 of the revised paper. Almost sure performance is achieved because we guarantee that the state is nonexplosive and hence the random fluctuations are averaged out as the time goes to infinity.
>
> - Experimental results
>
>   Thank you very much for the suggestion. We have added an ablation test of the safeguard mechanism in the simulation section.

---

> > ### Comment · Reviewer_ZdU6 · 2021-11-30
> > **Final Feedback**
> >
> > Thanks for the authors' responses and efforts on the revision, which help address some of my questions. To further improve the paper, I would suggest the following edits over the contents that still remain a bit unclear in the current version:
> >
> > - Notion of safety and safe learning: changing the term to “bounded cost” definitely makes it look better and more precisely reveals the nature of the objective in this paper. However, I am still concerned over the term “bounded-cost safe learning” and “bounded-cost safety”, as the common objective of safe learning and the definition of safety is clearly different from the focus here, which simply pursues J^{\pi_k} < +\infty for every k. To further clarify the goal in this paper, I would strongly suggest removing the term “safe” and replace with “bounded-cost”.
> >
> > - Contribution of the switch-based control strategy: the “almost sure” performance mainly relies on the piece-wise controller synthesized from the zero action with off-the-shelf innate stability guarantee and standard certainty equivalent controller (CE), which automatically gives the properties of bounded cost and reduced suboptimality gap (as shown through Theorem 1-3 and the related proof). However, as also pointed out by other reviewers, the “gap” between piecewise time-invariant linear system and time-varying nonlinear system due to such synthesis is not sufficiently investigated, e.g. if with small gaussian-like system noise, will the switch-based control strategy be as good as the standard CE? It would be more helpful if additional discussions could be provided with respect to the nonlinear switch-based policy as one piece instead of discussions over individual policies respectively. On the other hand, the usage of Markov parameter inference could be better motivated upfront, especially if itself presents an additional contribution besides the switch-based policy.

---

> > > ### Author Response · Authors · 2021-12-01
> > > **Response to final feedback**
> > >
> > > Thanks very much for the reviewer's appreciation of our revisions and valuable further comments.
> > >
> > > - About the term: we recognize that safe learning usually deals with constraint satisfaction, which is a different problem from ours. The motivation of defining "bounded-cost safety" of a learning process is that learning processes satisfying such property will prevent the state from exploding, and therefore safe to deploy in practice. We have searched our mind for more precise terms, but since the reviewer suggest that "bounded-cost safety" still causes confusions, we promise to change the term to "bounded-cost" in the further version.
> > >
> > > - Contribution of the switch-based control policy: we would like to state that the main contribution of our paper is the original switch-based strategy, together with proper coordination of the switching threshold and length of "non-action period", which guarantees a.s. convergence of both estimation error and policy suboptimality gap. Such guarantee was not present in previous works on online LQR. Also, the closed-loop system under our scheme is nonlinear due to the switching mechanism, whose analysis differs from existing works on learning LQR. We agree with the reviewer that the gap between "the nonlinear switch-based policy as one piece" and the optimal policy, i.e., the regret of the nonlinear switch-based policy, would be a stronger guarantee for the performance of the proposed scheme. That line of analysis, however, is still a challenging topic under investigation, and according to our current knowledge, we believe a complete presentation of which is a little beyond the capacity of this paper. That said, we will try our best to deliver interesting results in the further version.
> > >
> > > - Usage of Markov parameter inference: frankly the contribution of the inference part is relatively minor compared to the proposed design and analysis of switch-based policy, because the more common least-sqaures almost does the same job. However, we still believe this is an interesting alternative method that enrich the library of inference tools in the literature on online LQR, and we state the motivation of introducing such an alternative in Remark 3.
> > >
> > > Thanks again for your patience and valuable suggestions!

---

### Author Response · Authors · 2021-11-20
**Response to all reviewers**

We would like to express our gratitude to all the reviewers for their highly detailed comments and suggestions. We have addressed your concerns and updated the paper accordingly. The revisions are marked in red in the updated paper.

---

### Decision · Program_Chairs · 2022-01-20

**Decision:**

Reject

**Comment:**

The main contribution of the paper is in providing an additional layer to standard certainty-equivalent control for LQR dynamics, that essentially prevents the state from exploding exponentially, via forcing a "descent" to a small bounded state space in growing epoch sizes. Theoretically, this is shown to ensure a notion of boundedness which is termed as "bounded cost safety".

Overall, the reviewers raised several concerns in their initial reviews. These included the relevance of the proposed new approach to modeling "safety" here, whether it is actually novel given the assumptions about open loop stability of the original plant, the fact that the learning algorithm could take the state to arbitrarily large states durign the learning process, the significance of the contribution of the paper wherein a "kill switch" is effectively being designed depending on the state norm, whether least squares is an arguably simpler system estimation method compared to the impulse response estimator here, whether existing approaches based on an input failure probability parameter can be used to yield the same result by iteratively taking it down to zero, and other concerns about the technical exposition.

The author(s) provided detailed responses to the reviewers' concerns. Specifically, the safety/stability notion was clarified as being distinct from standard input-to-state stability, that the learning algorithm could, during its operation, drive the system state to arbitrarily large sizes (although asymptotically almost surely this is supposed to not happen), and how this paper's assumptions are different (and lighter) than other related work that achieves almost-sure guarantees.

In the discussion that ensued after the author response period, it was clear that the author responses had helped to address many of the reviewers' concerns. However, the overall impression has still not been convincing enough to recommend acceptance. This is primarily on two fronts, which I hope that the author(s) can address in the future to strengthen the submission: (1) The attempt at expressing "safety of learning" is found to be not satisfactory, in view of the observation that the proposed scheme does not guarantee classical notions of stability while in the process of learning. It also makes the main message of the paper confusing -- the reviewers noted that the words "safety" and "safe learning" still recur in the revised manuscript, creating scope for misinterpretation. (2) The algorithmic contribution appears to be incremental -- its essence is to "apply the brakes when the car runs too fast". This is not to take away from the hard work put in to analyzing the algorithm and deriving guarantees. (3) The experiments benchmarking the proposed strategy are not comprehensive
-- more relevant baselines drawn from existing work, such as the ones
that have emerged from the author response discussion, could be compared against. In fact, it would be compellingly in favor of this submission for the author(s) to show that other approaches fail to offer the same kind of "safe" performance that is expected.

Upon a more careful reading by myself in the recent past, I would also like to bring up a fundamental technical criticism about Theorem 1 and Defn. 2 ("bounded-cost safety"), which I believe must be fixed before the paper's conclusions can be accepted. Defn. 2 states that a learning process if bounded cost safe if for all times $k$, $\pi_k$ is not destabilizing in the sense that its value function is finite. However, the sequence of policies $\pi_k$, $k=1, 2, \ldots$ is a sequence of
*random* objects. So in what sense is Defn. 2 to be interpreted? If
the author(s) mean(s) to say that the event {$\forall k=1, 2, ...: \pi_k$ is not destabilizing} occurs w.p. 1, then this is a very strong requirement and cannot be guaranteed (random noise can cause a 'bad' controller to be learnt and applied at some time t with positive probability). Basically my contention is that Defn. 2 is incomplete for a theory-oriented paper like this, and as a consequence I do not see why Theorem 1 should hold (or more precisely, in what sense it should hold). The proof of Theorem 1 is not clear as well: The expectation in the third sentence of Sec A.2 ought to be taken for a
*fixed* controller $\hat{K}_k$ *always applied* to a system from time
zero until infinity. I do not see why a fixed controller's (standard infinite horizon average) cost should always be finite, leading me to suspect an irregularity in the proof argument. I wish this point could be discussed and resolved earlier in the author response phase, but it is unfortunately too late.